# Niobium: The Focus on Catalytic Application in the Conversion of Biomass and Biomass Derivatives

**DOI:** 10.3390/molecules28041527

**Published:** 2023-02-04

**Authors:** Luiz Oliveira, Márcio Pereira, Ana Pacheli Heitman, José Filho, Cinthia Oliveira, Maria Ziolek

**Affiliations:** 1Departamento de Química, Campus Pampulha, Universidade Federal de Minas Gerais, Belo Horizonte 31270-901, MG, Brazil; 2Instituto de Ciência, Engenharia e Tecnologia, Campus Mucuri, Universidade Federal dos Vales Jequitinhonha e Mucuri, Teófilo Otoni 39803-371, MG, Brazil; 3Faculty of Chemistry, Adam Mickiewicz University, Uniwersytetu Poznańskiego 8, 61-614 Poznań, Poland

**Keywords:** niobium compounds, biomass conversion, greenhouse gas

## Abstract

The world scenario regarding consumption and demand for products based on fossil fuels has demonstrated the imperative need to develop new technologies capable of using renewable resources. In this context, the use of biomass to obtain chemical intermediates and fuels has emerged as an important area of research in recent years, since it is a renewable source of carbon in great abundance. It has the benefit of not contributing to the additional emission of greenhouse gases since the CO_2_ released during the energy conversion process is consumed by it through photosynthesis. In the presented review, the authors provide an update of the literature in the field of biomass transformation with the use of niobium-containing catalysts, emphasizing the versatility of niobium compounds for the conversion of different types of biomass.

## 1. Introduction

### 1.1. Niobium: General Information

Niobium (Nb) is a chemical element with atomic number 41 and 92.90637(2) units of atomic mass [1,2] (Figure 1). It is a transition element belonging to group 5 of the periodic table of the elements, according to the IUPAC classification of the elements. Its name was inspired by Niobe, a Greek goddess daughter of Dione and Tantalus, whose name bears another element of group 5, tantalum. Niobium has physical and chemical properties similar to those of tantalum, making it difficult to distinguish between the two elements. In 1801, an English chemist, Charles Hatchett, described the discovery of an element similar to tantalum and named it columbium [3]. In 1809, William Hyde Wollaston (English chemist) mistakenly concluded that tantalum and columbium were identical [4]. A German chemist, Heinrich Rose, established in 1846 that the tantalum ores contained a second element which was baptized as niobium [5]. In 1864–1865, it became clear that “niobium” and “columbium” were two names used interchangeably for the same element. In 1949, the International Union of Chemistry officially recognized niobium as a chemical element [6].

Unlike common sense, niobium is not a rare element in nature; it is even more abundant than cobalt, lead, and tin (Figure 2). In nature, niobium is found in minerals such as columbite, pyrochlore, euxenite, struverite, ilmenorutile, ixiolite, loparite, and lueshite [7]. In the world, there are a few economically feasible niobium mines. Brazil is historically the world’s largest producer of niobium and ferroniobium (an alloy of niobium and iron), accounting for 75% of the niobium world’s production [8,9].

Niobium is the third most exported element in terms of value, losing only to iron ores and gold [11]. The world niobium consumption has grown a hundredfold since the 1960s, and the emergence of new technologies may contribute to increasing the niobium market in future. To maximize the exploitation of the full value of this natural wealth, we must learn to manufacture more sophisticated products with a consequent increase in its price. In 2010, the WikiLeaks website revealed a list of places vital to the United States of America, where the strategic importance of niobium was highlighted. Niobium can already be considered an essential input for the aerospace, oil and gas, and naval and automotive industries. In addition, various studies around the world are being carried out to use niobium compounds in different areas such as catalysis [12], medicine [13], materials technology [14], energy [15] and environmental remediation [16] The wide range of commercial and under investigation applications of niobium make it comparable to the vibranium from the Marvel stories. Thus, we must repeat what was achieved in Wakanda and explore the enormous potential of niobium to develop technologies that can be useful to society.

### 1.2. Different Areas of Niobium Application

The first commercial applications of niobium date back to the beginning of the 20th century when it was applied in metallurgy. Up to now, niobium has been used to make special steels due to its high resistance to corrosion and external temperatures (it melts at 2468 °C and evaporates at 4744 °C). Steels with greater tenacity and lightness can be produced by adding grams of niobium to a ton of steel. DeArdo [17] has provided an excellent review of the effects of niobium on the composition, processing, microstructure, and mechanical properties of different steels. Niobium has also been used to fabricate automobiles, air turbines, electronic devices, tomographs for magnetic resonance imaging, high-intensity lamps, gas pipelines, and optical lenses [18].

Niobium compounds such as NbTi and Nb3Sn have been used to solve problems in high-energy physics and the thermonuclear power industry. NbTi and Nb3Sn superconductors are constituents of the international thermonuclear experimental reactor (ITER) [19], large hadron collider (LHC) [20], TOKAMAK-like thermonuclear reactors, and fusion power demonstration reactors (DEMO) [21]. Nb3Sn has attracted attention due to its ability to carry current densities higher than the limits established for the commonly used NbTi. The effects of Sn content, morphology, and strain on the superconducting properties of Nb3Sn have been reviewed by Godeke [22] Popova and Deryagina [23] have recently reported the influence of the manufacturing methods, alloying, the geometry of wires, and heat treatment modes on the formation of Nb3Sn layers, structure, and morphology on the superconducting properties of Nb3Sn. More details on Nb-based superconductors can be found in recent reviews reported elsewhere [24,25].

In recent decades, niobium has been studied for applications in strategic areas such as energy [26], health [27] and catalysis [28]. As far as energy-related applications are concerned, different niobium compounds have been used in energy conversion and storage systems such as batteries [29], supercapacitors [30], solar fuel production [31,32] and solar cells [33,34]. Particularly, niobium-based oxides have been used as electrodes for batteries (Li-ion, Na-ion, vanadium redox flow), supercapacitors, and fuel cell applications [35] Charge storage mechanisms in different oxide structures and electrochemical activities regarding the specific capacitance/capacity, rate capability, and cycling stability of niobium-based oxides have been extensively described [36,37]. Deng et al. [15] have reported a comprehensive review on the progress of Nb-based oxides (Nb_2_O_5_, TiNb_x_O_2+2.5x_, M-Nb-O family) for advanced electrochemical energy storage applications (e.g., Li-ion batteries, Na-ion batteries, and Li/Na-ion hybrid supercapacitors).

In healthcare, niobium has been used in biomedical implants [38], antivirals [27], antibacterial products [39], tumor treatment [40,41], etc. Particularly, polyoxoniobates (PONs) can inhibit tumor development by changing the electron density of diseased cells and preventing their growth via oxidative stress [42]. Cytotoxicity tests show that the PONs inhibit tumor cell growth in a concentration-dependent manner. In addition, under UV irradiation, the tumor inhibition is enhanced, and the cytotoxic activity of PONs increases fivefold. The higher activity under illumination and low toxicity makes PONs attractive for use in photodynamic therapy. The higher cytotoxic activity under UV irradiation is due to the formation of reactive oxygen species (ROS) that can promote the selective oxidation of tumor cells. Compared to organic pharmaceuticals, PONs-based compounds are less costly and easier to synthesize, so their development can contribute to saving lives throughout the world.

The important role of niobium species in photocatalytic processes has been pointed out in several papers. The ROS formation via UV irradiation combined with H_2_O_2_ treatment of niobium pentoxide can be used in environmental protection for dyes removal from wastewater. Niobium pentoxide is a wide bandgap semiconductor that has been successfully used as a photocatalyst for the degradation of various organic compounds [43,44,45,46,47,48,49,50,51,52]. This metal oxide is also a promising catalyst for oxidation processes with the use of hydrogen peroxide as oxidant because it activates H_2_O_2_ to form ROS such as hydroxyl radicals (•OH), superoxide radical anions (O_2_•−) and peroxo species (O_2_^2−^), which are highly active in the oxidation of organic compounds [53,54,55,56]. The high activity of niobia in both photocatalytic and Fenton-like processes makes it possible to use niobium in hybrid water treatment technologies [50,57]. A combination of photocatalytic and Fenton-like processes over Nb_2_O_5_ allowed much higher efficiency of rhodamine discoloration than that observed for the less complex processes with the use of light or hydrogen peroxide only. Thus, the discovery of niobia properties opened a new pathway for the development of innovative catalytic systems, which is effective for environmentally friendly water treatments.

The photocatalytic properties of Nb_2_O_5_ have also been used in methanol oxidation, in which niobia-supported gold and copper were especially attractive [58]. The bimetallic Au-Cu/Nb_2_O_5_ catalyst was strongly affected by the nature of the light source: visible light promoted the total oxidation of methanol, whereas UV light was effective in a partial oxidation of this alcohol. Thus, depending on the light source, one can control the application of the niobia-supported gold–copper catalysts to the removal of alcohol from wastewater (environmental protection) or to the production of fine-chemicals (partial oxidation).

In addition to niobium pentoxide, other niobium compounds appear to be attractive for photocatalysis. NaNbO_3_, KNbO_3_, LiNbO_3_, and HNb_3_O_8_ have been used as photocatalysts for the CO_2_ conversion into methane [59,60]. Niobium-based perovskite oxynitrides, ANb(O,N)_3_ (A = Ca, Sr, Ba, and La), with narrow bandgap energies, are promising photocatalysts for the water-splitting reaction [61]. The use of BaNbO_2_N as a photocatalyst for water oxidation has been discussed in the review by Ahmed and Xinxin [62]. Niobium has also been used as a dopant to improve the catalytic activity of different materials for the oxidation of pollutants in water or in the synthesis of value-added molecules [56,63,64,65,66,67].

Although niobium compounds have been used in different areas, most of the studies concern heterogeneous catalysis in which they have been employed as catalysts, promoters, or catalyst support. Table 1 summarizes the review papers on the synthesis, characterization, and catalytic applications of niobium compounds over the last 30 years. In 1995, Tanabe and Okazaki [68] reviewed the catalytic properties of niobium in different reactions such as alcohol dehydration, olefin hydration, esterification, etc. In 1999, Nowak and Ziolek [69] reported a comprehensive review on the preparation, characterization, and application of niobium compounds in heterogeneous catalysis. In 2003, Ziolek [70] reviewed the oxidation reactions in liquid and gas-phase catalyzed by niobium-containing materials, and Tanabe [71] revised the catalytic applications of niobium compounds in the oxidative dehydrogenation of alkanes, the oxidative coupling of methane, oxidation and ammoxidation, and the removal of nitrogen oxides. In 2006, Andrade and Rocha [72] wrote a mini-review on the application of niobium compounds in organic synthesis. In 2009, Guerrero-Perez and Bañares [73] briefly reviewed the role of niobium in selective oxidation reaction (e.g., oxidative dehydrogenation of lower alkanes, oxidation of ethane to acetic acid, oxidation and ammoxidation of propane, partial oxidation of methane, oxidation of n-butane to maleic anhydride, and oxidative coupling of methane). In 2012, Nowak [74] reported in a review the use of niobium-containing molecular sieves in liquid and gas-phase oxidation reactions. Zhao et al. [75] described the methods used in the synthesis of Nb_2_O_5_ nanostructures for use in catalysis. The physical properties of niobium oxides (stoichiometric and non-stoichiometric) and niobates were well described by Nico et al. [76]. In 2017, Ziolek and Sobczak [12] reviewed the role of niobium-containing zeolites in the redox properties of Cu, Ag, Au, and Pt for cyclohexene, glycerol, and methanol oxidation reactions. Yan et al. [77] reported, in a perspective, the light olefin epoxidation by niobium catalysts. Moreover, in 2018, Wawrzynczak et al. [78] reviewed the catalytic behavior of niobium-containing mesoporous silicates and polyoxometalate for water splitting, epoxidation, basic catalysis, condensation, and the oxidation of unsaturated compound reactions. Lastly, in 2019, Siddiki et al. [79] reported a review on the use of Nb_2_O_5_ and Nb_2_O_5_.H_2_O in nucleophilic substitution reactions in the presence of basic inhibitors.

### 1.3. Scope of the Review

Over the past ten years, the use of niobium compounds in the catalytic conversion of biomass and their derivatives has increased significantly (Table 1). The conversion of glycerol, the by-product derived from those reactions, into value-added molecules will also be discussed. The proposed review fills the gap in the review papers devoted to niobium catalysts because it focuses on the catalytic reactions not covered yet in any review article. We hope this paper may be an inspiration for students and researchers from academia and industry to develop more efficient and selective niobium catalysts for converting plant biomass derivatives into products that may be useful to society.

## 2. Conversion of Plant Biomass Fractions over Niobium-Containing Catalysts

Among renewable resources, plant biomass generated by photosynthesis has an enormous potential to be catalytically converted into value-added chemicals and fuels. Vegetable biomass consists basically of C6- and C5-sugars that form cellulose (a linear polymer of D-glucopyranose connected by β-1,4-glycosidic bonds) and hemicellulose (a heteropolymer formed by different sugar monomers) (Figure 3). The third component, lignin, is an amorphous polymer consisting of phenylpropanol monomers joined by carbon–carbon and ether bonds [80,81]. Microcomponents such as fatty acids, tannins, and inorganic salts as well as other carbon- and hydrogen-rich substances (e.g., essential oils, vitamins, and dyes) may also be isolated from plant biomass [82,83]. In this section, the conversion of cellulose, hemicellulose, lignin, and fatty acids into different products catalyzed by niobium compounds is reviewed from 2010 to this date.

### 2.1. Cellulose and Hemicellulose Conversion to Different Products

C5- and C6-carbohydrate fractions that comprise hemicellulose and cellulose can be extracted from the plants by various pretreatments such as acid or alkali hydrolysis, ionic liquids, steam explosion, and further converted into molecules with greater commercial value [84]. Different niobium catalysts such as NbOPO_4_, Nb_2_O_5_, and HNbMoO_6_ have been used to convert cellulose into various molecules such as sorbitol, isosorbide, HMF, lactic acid, levulinic acid, ethylene glycol, and methyl levulinate (Figure 4). The performance of different niobium catalysts for cellulose conversion is summarized in Table 2.

#### 2.1.1. Sorbitol and Isosorbide

Sorbitol is a platform molecule used in the synthesis of fuels (e.g., ethanol, hexane) and molecules used in the food, cosmetics, medical and paper industries. Sorbitol can be synthesized from cellulose hydrolysis to glucose followed by glucose hydrogenation into sorbitol (Figure 5). The catalytic hydrolysis of cellulose is performed by acid catalysts and hydrogenation by metal catalysts. Afterwards, sorbitol is converted to sorbitan isomers by dehydration. Subsequent sorbitan dehydration leads to the formation of isosorbene (Figure 5).

Xi et al. [85] used a Ru/NbOPO_4_ bifunctional catalyst for the selective conversion of cellulose to sorbitol. The acidic sites of mesoporous NbOPO_4_ promote the hydrolysis of cellulose to glucose and Ru nanoparticles hydrogenate glucose to sorbitol. Thanks to the dual functionality of Ru/NbOPO_4_ catalyst, sorbitol was produced in 59–69% yield at above 90% cellulose conversion. Successive dehydrations of sorbitol and sorbitan give isosorbide (Figure 5).

Isosorbide is a platform molecule used as an intermediate in the pharmaceutical industry, an additive to give strength and rigidity to polymers, and a monomer for the biodegradable polymer [100]. Isosorbide is currently produced from edible starch by Roquette company, which is one of the largest producers of this compound. Thus, the development of synthetic routes using inedible raw materials is desirable to avoid competition with the food industry. It has been indicated that isosorbide can be produced from cellulosic biomass. Xi et al. [101] produced isosorbide in approximately 57% yield using Ru/NbOPO_4_ catalysts to depolymerize cellulose through hydrolysis and hydrogenation. Then, NbOPO_4_ with a large amount of acid sites was used to obtain isosorbide from sorbitol and sorbitan dehydration.

#### 2.1.2. 5-Hydroxymethylfurfural

5-Hydroxymethylfurfural (HMF) is a value-added platform molecule that can be obtained from acid hydrolysis of cellulose into glucose and fructose, which is followed by carbohydrate dehydration (Figure 6) [102].

The conversion of cellulose to HMF has been carried out in water or biphasic solvent mixtures (e.g., THF/H_2_O, MIBK/H_2_O), at temperatures of 140–200 °C, using niobium catalysts such as Nb_2_O_5_/C [91], Nb_2_O_5_ [89] NbOPO_4_ [87], NbOx/ZrO_2_ [88] and Nb-SBA-15 [90] HMF yields obtained with NbOPO_4_, Nb_2_O_5_, and NbOx/ZrO_2_ were about 16–19%. Using Nb-SBA15 and Nb_2_O_5_/C, yields greater than 50% and conversion above 90% can be achieved.

HMF can be converted to various molecules such as 5-alkoxymethylfurfural, 2,5-furandicarboxylic acid, 5-hydroxymethylfuroic acid, 2,5-bis-hydroxymethyl furan, 2,5-dimethylfuran, bis (5-methylfurfuryl) ether, levulinic acid, adipic acid, 1,6-hexanediol, caprolactone, and caprolactam, which have high potential to be used as fuel or polymers (Figure 7) [103].

It is worth mentioning that HMF is considered as an extremely important platform molecule for lignocellulose biorefinery due to the high industrial applicability of its downstream products such as furan-2,5-dicarboxylic acid (FDCA), which is known as the green substitute for terephthalic acid in the manufacture of polymers [103,104].

Fergani et al. [105] have synthesized FDCA from HMF using Nb-zeolites and found that the greater the ratio of basic to acidic sites, the higher the HMF conversion as well as selectivity for FDCA. The acidic characteristics of Nb_2_O_5_ play a significant role even when applied as support. Duan et al. [106] have studied the transformation of HMF into 2-hydroxy-3-methyl-2-cyclopenten-1-one (MCP, a compound widely used as flavoring agent). This reaction occurs in two stages: (i) hydrogenation of HMF in 1-hydroxy-2,5-hexanedione (HHD) over Pd/Nb_2_O_5_ and (ii) isomerization (through aldol condensation) over basic catalysts. According to the authors, the acidity of Nb_2_O_5_ support, detected by temperature-programmed desorption analyses, was crucial for increasing the HHD selectivity.

#### 2.1.3. Levulinic Acid

Acids derived from biomass are strategic molecules because they are easily converted to esters by acid or basic catalysis. If these esters have a long chain, a structure similar to biodiesel, they can also be applied as fuels, since their oxygen content is reduced, increasing their calorific value during combustion [107]. Among the acids resulting from the conversion of lignocellulosic matter, the most important is levulinic acid (LA) [108], which is confirmed by a large number of publications on this compound. Exploring the Lewis acid potential of transition metal salts for the esterification of LA with 1-hexanol, Jia et al. [107] have achieved a conversion of LA close to 90% and yield of above 60% with the use of NbCl_5_. This catalyst showed one of the best performances, being better than ZnCl_2_, CuCl_2_, SnCl_4_·5H_2_O, CrCl_3_·6H_2_O, VCl_3_, FeCl_3_, AlCl_3_, MoCl_5_, ZrCl_4_, HfCl_4_, and H_2_SO_4_, equivalent to TaCl_5_ and worse only than WCl_6_.

From another application perspective, the short-chain esters produced from LA can be applied as flavorings, fuel additives, polymer precursors, and green solvents. Another LA-derived compound that has attracted much attention is γ-valerolactone (GVL) [109]. GVL is produced by the reduction of the LA carbonyl group (generating the γ-hydroxyvaleric acid, GHV) followed by intraesterification to form a five-membered ring [110]. The reduction step determines the reaction rate. Different approaches such as hydrogenation, catalytic hydrogen transfer, and photocatalysis have been used to reduce LA to GHV [109]. Each method has its peculiar features, but the photocatalysis and catalytic hydrogen transfer procedures have been proposed to avoid the use of gas hydrogen, whose presence makes the process more dangerous and less sustainable (since most of the hydrogen produced comes from non-renewable sources).

Levulinic acid (LA) is also a molecule used to synthesize different molecules as polymer precursors, pharmaceutical intermediates, and fuel additives (Figure 8). LA can be produced from cellulosic biomass by a chemical route from HMF. At first, cellulose undergoes depolymerization by acid hydrolysis to release sugars that are dehydrated into HMF. The subsequent dehydration of HMF leads to levulinic acid (Figure 9). Ding et al. [111,112] have obtained a 53% levulinic acid yield from cellulose using Al-NbOPO_4_ as a catalyst. They have found that the Al doping increases the number of Lewis and Bronsted acid sites in NbOPO_4_. In particular, an increase in the number of Lewis acid sites favors the formation of LA from cellulose.

#### 2.1.4. Methyl Levulinate

On an industrial scale, levulinate esters are produced by homogeneous catalysis through the esterification of levulinic acid with alkyl alcohols in an acidic medium. Alternatively, levulinate esters may be directly synthesized from cellulose by heterogeneous acid catalysis with alcohol (Figure 10) [113]. In the presence of methanol, cellulose is sequentially converted to methyl glycoside, 5-methoxymethylfurfural, and methyl levulinate. Using NbOPO_4_ as a catalyst and methanol as a solvent, about 98% cellulose conversion and 58% selectivity for methyl levulinate can be achieved at 180 °C in 24 h.

#### 2.1.5. Hexane

The production of liquid alkanes and cycloalkanes from lignocellulosic substrates is an extremely challenging and highly coveted route, since it is a sustainable alternative to the use of non-renewable sources (such as petroleum and coal). Xia et al. [98] have demonstrated that this is feasible, with great mass and carbon yield, through a one-pot hydrodeoxygenation reaction using a Pt-loaded NbOPO_4_ catalyst. According to the authors, the synergy between the multifunctionality of the catalyst (Pt, NbOx species, and the acidic sites) allowed a high-performance hydrodeoxygenation, efficiently converting the bulk lignocellulose into liquid alkanes: pentane, hexane, and alkylcyclohexanes (Figure 11).

#### 2.1.6. Lactic Acid

Lactic acid has been widely used in the formulation of cosmetics, production of biodegradable polymers, and food additives [92]. The industrial production of lactic acid occurs through the anaerobic fermentation of six-carbon sugars such as glucose [93] derived from lignocellulosic biomass. Although the fermentation processes are efficient, they require ultra-specific conditions to function properly such as low O_2_ concentration, adequate pH, and the absence of contaminating microorganisms. This limits scale expansion and reflects directly on the high market price of lactic acid.

Alternatively, lactic acid can be produced from the catalytic conversion of cellulose, its monomer (glucose), and dihydroxyacetone, a three-carbon monosaccharide that can be obtained from the oxidation of glycerol [92,93,114,115]. In this context, the Nb-based catalysts are exceptional for such purposes thanks to the strength of their acidity, the balance between acidic sites (Bronsted and Lewis) and their stability in different reaction media [116,117].

#### 2.1.7. Ethylene Glycol

Ethylene glycol is a diol that has been mainly used as an antifreeze agent and as a monomer in the manufacture of polyester. Its synthesis is based on the selective oxidation of petroleum-derived ethylene, generating ethylene oxide, which is subsequently hydrated, generating ethylene glycol (Figure 12). However, efforts have been made to obtain this molecule from an important component of lignocellulosic biomass, cellulose. The presence of niobium in the catalyst structure favors both cellulose conversion and ethylene glycol selectivity, regardless of whether this element is being used in small quantities (dopant) or in bulk form (metal oxide) [86,96]. It has been demonstrated that niobium phosphate can facilitate the C-C bond cleavage in the biopolymer, favoring the subsequent hydrogenation by Ru to form ethylene glycol [86].

#### 2.1.8. Water Soluble Sugars

The first step toward the enhancement of lignocellulosic biomass is the depolymerization of its fundamental units: lignin, hemicellulose, and cellulose. Among them, cellulose has attracted much attention because it has the least structural complexity and holds the largest mass fraction in lignocellulosic material compared to the other two components. Furthermore, this biopolymer can be cleaved into oligomers, dimers, or even glucose monomers, and the latter is easily converted into downstream products. Despite the interest in cellulose, its depolymerization is not an easy task because of its crystalline and poorly flexible structure due to the β-1,4 linkage between the glucose monomers and the high density of hydrogen bonds along the chains [28]. Through the mechanocatalytic technique employing a layered niobium molybdate catalyst (HNbMoO_6_), Furusato et al. [97] have been able to depolymerize cellulose into water-soluble sugars using a ball mill in the absence of any solvent. The yield of the products was 14% (oligosaccharides, cellobiose, cellotriose, cellotetraose, cellopentaose and cellohexaose, as well as its epimer mannose and anhydrous sugars such as glucose and cellobiose). The excellent depolymerization performance was attributed to the strong acidity and layered structure of HNbMoO_6_.

### 2.2. Lignin Conversion

The final objective of the lignin valorization is to be able to cleave the biopolymer (Figure 3) to obtain monoaromatics that, in turn, already have been applied mainly in the polymeric industry. Lignin fragmentation occurred quite efficiently in the presence of Co@Nb_2_O_5_@Fe_3_O_4_ or Fe_3_O_4_@Nb_2_O_5_@Co@Re catalysts [95,116]. The complete transformation of lignocellulosic biomass into platform molecules, such as 5-hydroxymethylfurfural (HMF) and furfural, is of great importance. Ru/Nb_2_O_5_ catalysts are the most promising from among those reported in the scientific literature [118,119,120]. The selective hydrodeoxygenation of aromatics and alkanes without oxygen has been achieved in nickel nanoparticles supported on Zr-doped niobium phosphate acid catalysts in an autoclave, under mild conditions. Zr-doped nickel in niobium phosphate exhibited excellent activity, converting 84% of diaryl ethers and showing better selectivity toward oxygen-free aromatics, of around 62% [120]. Other ways of converting lignin are studied using niobium containing catalysts [121,122,123,124,125]. Table 3 summarizes the main niobium-based catalysts used for the lignin conversion.

### 2.3. Fatty Acids Conversion to Different Products

Triglycerides in vegetable oils are hydrolyzed into long-chain fatty acids and glycerol in the presence of acidic or basic catalysis. The resulting acids are then converted into esters in the reaction known as transesterification (Figure 13). Methanol and ethanol are the main alcohols used as nucleophiles, and the compounds produced in this reaction are green fuels called biodiesel. The industrial production of biodiesel is carried out using homogeneous catalysis, which makes the process less sustainable [130]. The use of heterogeneous catalysis has been pointed to as an alternative to improve the “environmentally friendly” character of the process because this route facilitates the recovery of the catalysts after reaction, reduces the generation of residues formed during the washing step of the biodiesel and allows the reuse of the solid catalyst in several batches. Different heterogeneous acid catalysts have been tested for this purpose, including niobium compounds that have proved successful for the esterification of biomass fatty acids, especially due to their strong acidic properties. Gonçalves et al. have discovered by experiment design that esterification with methanol catalyzed by niobic acid is favored for a low unsaturation degree and short chain of the fatty acid [131].

Surprisingly, niobium catalysts can go beyond the esterification, transforming triglycerides into alkanes [132]. This feat was performed using a Pd/NbOPO_4_ catalyst in the hydrodeoxygenation of vegetable oils at low temperature. The conditions were perfect to prevent the exacerbated cleavage of the carbon chains, resulting in a high selectivity of n-alkanes and propane (the latter produced from the glycerol fraction present in triglyceride). Once again, the synergistic effect of niobium with noble metals was observed; however, this activity does not rely strictly on the presence of noble elements. Some studies indicate that the bare niobium phosphate can transform triglycerides of soybean oil into biogasoline, green diesel, and bio-jet fuel, in addition to other hydrocarbons in the presence or absence of H_2_ [133,134].

## 3. Conversion of Plant Biomass Derivatives over Niobium-Containing Catalysts

### 3.1. Glucose Conversion to Different Products

Glucose is formed by the hydrolysis of cellulose, which is the main component of lignocellulose (about 30–50%). This simple sugar has a unique stereostructure, and therefore, methodologies for the effective catalytic conversion of glucose to useful chemicals, while maintaining the original stereostructure, are of great interest in order to make the best use of biomass. The scheme presented in Figure 14 shows different ways of glucose transformation: (i) the pathways without the C-C bond cleavage (transformation of glucose to succinic acid, gluconic acid and 5-hydroxymethyl furfural (HMF)) and (ii) with the C-C bond cleavage (ethanol and lactic acid formation). Each catalytic conversion of glucose involves different active centers on the catalyst surface of different strength, and most of them are able to provide niobium species in various environments.

HMF is the most important chemical produced from glucose, as it is widely applied as a platform for the production of liquid fuels and valuable fine chemicals. The catalytic transformation of HMF is described in Section 2.1.2. HMF can be easily obtained from fructose, but the cost of fructose is very high because the isomerization of glucose to fructose requires an expensive chromatographic step [90]. Moreover, fructose is not a stable compound. Therefore, the focus is on a one-step catalytic transformation of glucose to HMF. The conversion of glucose to HMF requires tandem steps (Figure 15); in the first one, glucose isomerization occurs in the presence of bases, enzymes or Lewis acid centers (LAS) [90], while Brønsted acids (BAS) are involved in the second step of fructose dehydration to HMF. The co-existence of BAS and LAS is necessary for the efficient conversion of glucose to HMF in a one-pot catalytic system. Thus, much effort has been made to search for the catalysts containing both BAS and LAS, which would be able to activate both steps of the tandem reaction. From among such catalysts of special importance are the niobium-containing materials, because niobium species in different environments can act as BAS, LAS or both types of acidic sites. Niobium catalysts described in the literature for glucose transformation to HMF are summarized in Table 4.

Bulk niobium (V) oxide hydrate has both LAS and BAS on its surface. Carlini et al. [136] have found that niobic acid was a very promising catalyst for the dehydration of glucose in a THF/H_2_O biphasic reaction system. However, a later study by Carniti et al. [137] indicated that it was ineffective for glucose conversion to HMF because of the presence of strong acidic centers close to one another, which leads to the formation of humines and/or other insoluble residues on the Nb_2_O_5_ surface and to fast deactivation of the catalyst. It has been evidenced that the dispersion of niobia in/on silica matrix permits maintaining the catalytic activity in the dehydration of fructose (the second step in glucose transformation to HMF) in water for a long time without any deactivation compared to the fast deactivation observed on bulk niobium oxide hydrate. Taking this into consideration, Peng et al. [90] have synthesized the ordered mesoporous molecular sieves, Nb-SBA-15, in which niobium species were dispersed in a silica matrix. In this material, NbO_4_ units of the hydrated surface niobia species on silica are connected by Nb-OH-Nb bridges playing a role of BAS, whereas the niobium species in mononuclear tetrahedral NbO_4_ located in the silica framework are weak Lewis acid centers due to the charge transfer between the tetrahedral oxygen ligands and the central niobium ions. It has been documented that the material containing Si/Nb at the molar ratio equal to 40 exhibited a suitable number of LAS along with the right ratios of BAS/LAS and strong LAS/total LAS. The two centers of the two types interacted with each other and provided the highest HMF selectivity and yields as well as stability of the catalysts in ten recycling runs. With increasing niobium content, the ratio of strong LAS/total LAS increased and the selectivity to HMF decreased.

Instead of using a silica matrix for niobium, aluminosilicates can be applied for obtaining attractive catalysts for glucose conversion to HMF. In such catalysts, the additional source of acidity is aluminum species. Silica-alumina composite (AlSiOx) is known to efficiently catalyze the glucose conversion to HMF, but doping with niobium species has been found to increase the yield of HMF [137]. A high selectivity to HMF requires weak Lewis acid sites on the catalyst surface because the medium and strong LAS are responsible for the undesired by-products (lactic acid, humins) formation. Moreover, the BAS/LAS ratio, dependent on the amount of niobium dopant, influenced the HMF selectivity, and the character of this relationship was described by the volcano curve. The highest selectivity to HMF (71%) with 92.6% of glucose conversion was achieved over the Nb_0.15_Al_0.85_Si_25_O catalyst. Moreover, niobium-doped mesoporous AlSiOx catalysts showed good recycle performance.

Niobium-doped zeolites were also applied for the glucose conversion to HMF as bifunctional catalysts [110]. Zeolite Beta used as a matrix for niobium doping was dealuminated before niobium loading, and therefore, the amount of aluminum in Nb-Beta was lower than in the AlSiO system. The mesoporous Nb-Beta zeolites contained residual framework Al acid sites, extra-framework isolated Nb(V) and Nb_2_O_5_ pore-encapsulated clusters in which Nb(V)O-H exhibited moderate strength Brønsted acidity. The presence of Nb(V)O-H species with niobium linked by Nb-O-Si bonds to the zeolitic walls increased the acid sites’ concentration. Moreover, the presence of these species also stabilized the zeolite, which was most probably by polarizing the framework. Thanks to the high stability, the catalyst could be recovered several times by centrifugation and regenerated by simple calcination. It was indicated that the niobium doped beta zeolite was more active and selective to HMF (84.3% selectivity to HMF for a glucose conversion of 97.4%) than the dealuminated beta zeolite studied by Otomo et al. [138] (84.3% selectivity to HMF for a glucose conversion of 97.4%).

Niobia dispersed on carbon supports [91,139] and on tungsten oxides [140,141,142,143] were also used as catalysts for glucose transformation to HMF. Yue et al. [140] in a detailed study of Nb-WO_3_ and P/Nb-WO_3_ systems (in the latter, the number of BAS was reduced by phosphate doping) have found that fructose, obtained as a result of the isomerization of glucose in the first step of the reaction, was not directly dehydrated to HMF but intermediate products were formed first (as shown in Figure 16). The rate-determining step in the HMF production was dehydration of the intermediate product. The Nb-WO_3_ catalysts in which Nb was incorporated in a tungsten oxide structure had much lower Brønsted acidity than Nb_2_O_5_, and consequently, the favored by-products were formed during glucose conversion. The isomerization and dehydration occurred at a high rate on the LAS and weak BAS of WO_3_ and Nb-WO_3_ catalysts. The by-products were very fast transformed to HMF by the addition of mineral acid because the last dehydration step to HMF requires stronger BAS. DFT calculations indicated that the tungstite surface can adsorb glucose, abstract protons from glucose and catalyze the hydride shift to fructose, whereas Nb substituted to a WO_3_ structure stabilizes the deprotonated glucose intermediate and lowers the overall activation barrier for glucose isomerization.

Wiesfeld et al. [141] have also used WO3 doped with niobium species and found that the doping of tungstite with titanium or niobium improved its activity by optimizing the ratio between Lewis and Brønsted acid sites. They identified the active centers in the M-WO_3_ systems (M = Nb or Ti) as illustrated in Figure 17. Both niobium and titanium led to increased Brønsted acid sites densities in the tungstite phase, which had a positive effect on glucose dehydration, although the final 5-HMF selectivities were mostly unaffected.

Mixed mesoporous tungstate and niobium oxides were also prepared using a greater amount of niobium than tungstate or the same amounts of both of them [142,143]. In such systems, the number of Brønsted acid sites increased, while that of Lewis acid sites decreased, which favored the transformation of glucose to HMF. The dehydration of glucose in aqueous medium was achieved by tandem reactions including the isomerization of glucose to fructose over Lewis acid sites and fructose dehydration to HMF over Brønsted acid sites. The selectivity to HMF was improved with a 2-butanol/H_2_O system [142]. The organic phase may extract the formed HMF from the aqueous phase, thus suppressing HMF rehydration and condensation, leading to the formation of by-products (e.g., levulininc acid and formic acid—products of HMF rehydration).

Niobia dispersed on carbon supports was also applied in glucose conversion to HMF [91,140]. Unfortunately, the drawback of the system is that the carbon-supported catalysts tend to locate in the organic phase due to their hydrophobic nature, while the reactant glucose is present in the aqueous phase. Therefore, carbon support has to be functionalized to increase the degree of its hydrophilicity. The effect of the location of the niobia/carbon composites on the catalytic performance has been studied in the conversion of D-glucose to HMF in a biphasic system [140]. It was found that the niobia catalyst located in the aqueous phase showed the highest glucose conversion.

Another group of catalysts studied for possible use in the one-pot synthesis of HMF from glucose is based on niobium phosphate, NbOPO_4_, which could be produced in such a manner that the BAS/LAS ratio would be optimal for conversion of glucose to HMF [87,115,143] and niobia doped with phosphorous acid [144,145]. The importance of the BAS/LAS ratio and the possibility of its control was pointed out [143]. The authors of this report have disclosed that an excess of Lewis acidity on the catalyst surface leads to unselective glucose transformation into humins. Modification of the catalysts by a silylation procedure or by deeper treatment with phosphoric acid led to a significant increase in the selectivity to HMF due to the deactivation of unselective Lewis acid sites. In view of the above, it was proposed that a synergism of a protonated phosphate group and a nearby metal Lewis acid site in the two-stage glucose transformation into HMF permitted a highly selective glucose dehydration. Another possibility of BAS/LAS control has been shown by Catrinck et al. [146]. The authors indicated that when a mixture of niobia and niobium phosphate was used as a catalyst, the achieved combination of Brønsted and Lewis acidity was more suitable for an effective glucose dehydration to HMF than in the presence of each individual catalyst. The acidity of NbOPO_4_ was also modified with a surfactant (CTAB—cetyltrimethylammonium bromide) [115]. Interestingly, it was demonstrated that NbOPO_4_/CTAB exhibited a much higher HMF yield in glucose conversion than in fructose dehydration to HMF, although fructose is commonly recognized as an intermediate product in glucose conversion to HMF.

Niobium-containing catalysts can be also applied in the transformation of glucose to carboxylic acids. For this purpose, usually, bifunctional catalysts are required. Fergani et al. [105] have prepared beta zeolite with highly dispersed niobium species and successfully applied it in glucose oxidation to succinic acid. The catalyst exhibited very high stability against water used as a solvent. During preparation of the catalyst, aluminum was extracted from the zeolite framework and deposited as Al-O extra-framework species. In such a way, a bifunctional catalyst was formed. The Al-O species exhibited strong acidity, and the authors hypothesized their involvement in the dehydration of glucose to levulinic acid (LA) in the first step of the reaction. Next, LA was efficiently oxidized to succinic acid (SA). As the production of SA is a two-step reaction and it was performed in one pot, 12 h reaction time was needed to achieve 100% glucose conversion (at 180 °C).

HBeta zeolite with dispersed niobium species was also applied in the oxidation of glucose to gluconic acid [146] The authors used a different preparation procedure than that applied in [105], and the Nb/HBeta zeolite used in their work did not contain extra-framework Al-O species. The reaction was carried out only for 2 h at 110 °C, and the conversion of glucose was very low (7%). Nb/HBeta was also used as a support for gold, but it appeared that niobium dopant, although it acted as a structural promoter enhancing gold dispersion, considerably decreased the activity of the gold catalyst because of a very strong gold–niobium interaction.

The catalytic transformation of glucose to lactic acid requires the presence of Lewis acid sites on the catalyst surface and a reduced amount of BAS, because the presence of BAS together with LAS directs the reaction toward the conversion of glucose to HMF, as shown above. Lewis acid sites are favorable for the isomerization of glucose to fructose (the transformation of aldoses to ketose) and then retro-aldol condensation to the corresponding trioses (GLY and DHA), which was considered as the key step during the lactic acid formation from glucose [147,148]. Thus, the preparation of niobia catalysts by Cao et al. [114] was directed to the formation of niobia containing mostly Lewis acidity. These authors produced a Nb_2_O_5_ nanorod catalyst by a soft chemical process under hydrothermal conditions and used it for the catalytic conversion of glucose to lactic acid. Such a catalyst exhibited a remarkable catalytic potential in lactic acid production from glucose because of its high Lewis acidity in comparison with that of the conventional Nb_2_O_5_ nanoparticles. The authors obtained a ca. 39% yield of lactic acid in optimum conditions.

Niobium-containing catalysts are also involved in mannose production from glucose. Mannose is a six-carbon monosaccharide that makes up hemicellulose. Due to its benefits to human health, this sugar has been widely used in the formulations of dietary supplements and drugs [128]. Proportionally, the amount of mannose in the lignocellulose biomass is small compared to those of other sugars [149]. However, mannose can be synthesized by glucose epimerization. The latter process consists of the structural rearrangement of saccharide by modifying one of the chiral centers that consequently generates the diastereoisomer of the starting molecule. D-mannose is a C2 epimer of D-glucose: that is, the difference between them is the inversion in the configuration of the carbon adjacent to the aldehyde (linear form) or hemiacetal (cyclic form) (Figure 18). Takagaki et al. [150] have epimerized with high selectivity D-glucose to D-mannose using niobium molybdates, which was independent of acidity. Similar results (see Table 4) were observed for the strongly acidic (HNbMoO_6_) and non-acidic (LiNbMoO_6_) catalysts.

**Table 4 molecules-28-01527-t004:** Performance of niobium-based catalysts for the conversion of glucose into different products—from 2010 to date.

Entry	Catalyst	Solvent	t (h)	T (°C)	Product	C (%)	S (%)	Y (%)	Experimental Conditions	Ref.
1	Fe-doped niobium phosphate	Water	3	180	Levulinic acid	99	65	64	0.056 M glucose; 0.050 g catalyst.	[151]
2	Nb_2_O_5_ nanorods	Water	4	250	Lactic acid	100	39	39	0.05 g catalyst; 0.10 g glucose; 20 g water; 800 rpm.	[114]
3	Nb_2_O_5_-Beta zeolite	Water	12	180	Succinic acid	100	84	84	p_O2_ = 18 bar; 0.05 g catalyst; 0.09 g glucose; 10 mL water.	[105]
4	H_3_PO_4_/Nb_2_O_5_·H_2_O	Water	3	120	HMF	92	52	48	0.20 g catalyst; 0.02 g glucose; 2 mL water.	[145]
5	Nb-doped Al,Si oxides	Water/THF	2	160	HMF	93	71	66	0.20 g catalyst; 0.20 g glucose; 2 mL water saturated with NaCl; 6mL THF.	[137]
6	Nb-modified Beta-zeolite	Water/MIBK	2	180	HMF	97	84	81	0.03 g catalyst; 0.18 g glucose; 3.5 mL water saturated with 20% NaCl and 1.5 mL methylisobutylketone (MIBK).	[138]
7	Nb-SBA-15 (Si/Nb ratio = 40)	Water/THF	3	165	HMF	94	66	62	0.10 g catalyst; 0.10 g glucose; 2 mL water saturated with NaCl; 6 mL THF.	[90]
8	Nb_2_O_5_·H_2_O treated with H_3_PO_4_ and heated at 300 °C	Water/2-butanol	2.3	160	HMF	86	63	54	0.10 g catalyst; 1.20 g glucose; 20 mL water, 30 mL 2-butanol; 800 rpm.	[145]
9	Nb_0.2_-WO_3_	Water/1-butanol	3	120	HMF	98	53	52	0.10 g catalyst; 0.01 g glucose; 1 mL solvent; 1-BuOH/H_2_O = 3.	[140]
10	Nb-doped WO_3_	Water/THF	4	120	HMF	100	55	55	0.04 g catalyst; 0.04 g glucose; THF/H_2_O volume ratio = 9.	[141]
11	Nb,W oxides	Water/2-butanol	2	140	HMF	100	52	52	0.20 g catalyst; 1 wt% glucose; 2-butanol/H_2_O volume ratio = 2.5.	[142]
12	Nb,W oxides	Water	2	120	HMF	36	53	19	0.20 g catalyst; 4.5 wt% glucose in water.	[143]
13	NbOPO_4_/CTAB	Water	1	140	HMF	87	44	38	0.03 g catalyst; 0.06 g glucose; 3 mL water.	[115]
14	Nb_2_O_5_·H_2_O/NbOPO_4_	Water	2	152	HMF	55	56	31	2 wt.% glucose in water.	[146]
15	NbOPO_4_	Water/MIBK	7.5	135	HMF	60	60	36	2 g catalyst; 4 g glucose in 20 mL and 3 vol. methylisobutylketone (MIBK). 500 rpm.	[143]
16	NbOPO_4_	Water	1	140	HMF	41	94	38	0.5 g catalyst; 0.1 g glucose in 10 mL water.	[87]
17	Niobia/carbon composites	Water/THF	4	160	HMF	98	60	59	0.10 g catalyst; 0.10 g glucose; 2 mL water saturated with NaCl, 6 mL THF.	[91]
18	Niobia/carbon composites	Water/SBP	2	170	HMF	78	26	20	0.10 g catalyst; 5 wt% glucose in water saturated with NaCl, sec-butyl phenol (SBP)/water mass ratio = 2.	[139]
19	LaOCl/Nb_2_O_5_	DMSO	3	180	HMF	65	82	53	0.10 g catalyst; 0.046 M glucose; 1000 rpm.	[152]
20	HNbMoO_6_	Water	1.5	120	D-mannose	33	88	29	0.01 g catalyst; 0.30 g glucose; 3 mL water.	[150]
21	LiNbMoO_6_	Water	1.5	120	D-mannose	26	91	24	0.01 g catalyst; 0.30 g glucose; 3 mL water.	[150]
22	NbOPO_4_-supported MgO	Water	0.5	120	Fructose	35	70	25	1 wt.% glucose. Frutose productivity = 13.6 g g^−1^_catalyst_ h^−1^.	[153]

### 3.2. Fructose Conversion to 5-methylhydroxyfurfural

Fructose is not a monosaccharide from the structure of lignocellulosic materials; it is an intermediate in the conversion of glucose into HMF. In this reaction, glucose is first isomerized (catalyzed by Lewis acidic sites) generating fructose that after dehydration (catalyzed by Bronsted acidic sites) produces HMF [77]. Niobium oxides, as previously mentioned, have excellent acidic properties, are chemically resistant and their ratio of Bronsted/Lewis acid sites can be tunable, which makes this solid an excellent candidate for the production of HMF from both glucose and fructose. Some of the great results available in the literature are presented in Table 4 (conversion of glucose to HMF) and Table 5 (conversion of fructose to HMF).

A large part of intermediates important for the chemical industry can be produced through sugars derived from biomass (Figure 6). Carbohydrate dehydration leads to the formation of furanic compounds such as furan-2-carboxaldehyde (furfural) and 5-hydroxymethyl-2-furanocarboxaldehyde or 5-hydroxymethylfurfural (5-HMF), which are key intermediates to synthesize various products of interest [153,154,155]. 5-HMF is a compound consisting of a furan ring containing an aldehyde group and a hydroxymethyl group in positions 2 and 5 [156,157]. HMF can be obtained through three dehydration reactions of biomass-derived C-6 carbohydrates and can act as a green chemical intermediate to synthesize a wide range of compounds [157]. Mesoporous niobium phosphate is an excellent solid acid for the dehydration of fructose to 5-hydroxymethylfurfural in water. The excellent catalytic activity obtained in the aqueous phase can be attributed to its high acid site density and the tolerance to water [157,158,159]. The catalytic dehydration of carbohydrates on exfoliatable layered niobic acid in an aqueous system under microwave radiation achieved a 5-hydroxymethylfurfural (HMF) yield of 55.9 % at a high substrate/catalyst weight ratio of 50 and 10 wt% fructose solution, which is close to the yield achieved by homogeneous aqueous systems [159]. Prado et al. [160] prepared a catalyst from the niobium filter cake which led to the conversion of fructose to HMF with a yield of 22% in aqueous medium and 47% in DMSO. Qiu et al. [161] have observed that the conversion of fructose to HMF is favored in the presence of polar aprotic solvents. A high yield of 96.7% with 100% fructose conversion was obtained using a niobium phosphotungstates catalyst in DMSO at 80 °C for 90 min. A niobium compound synthesized from the reaction of niobium chloride and nitrilotris (methylenephosphonic acid) (Nb–NTMPA) catalyzed the dehydration of fructose with a HMF yield of 85.6% in a mixture of mixture of N,N-dimethylacetamide (DMA) and NaBr [161]. Niobium oxide/mesoporous carbon composites were efficient to convert 100% fructose to HMF with a selectivity of 76.5% [162]. Niobium-doped TiO_2_ solid acid catalysts showed strengthened interfacial polarization, amplified microwave heating, and enhanced energy efficiency for hydroxymethylfurfural production. Nb doping has been demonstrated effectively enhance the quantity of acid sites and increase the acid strength [163].

**Table 5 molecules-28-01527-t005:** Performance of niobium-based catalysts for dehydration of fructose into HMF—from 2010 to date.

Entry	Catalyst	Solvent	t (h)	T (°C)	C (%)	S (%)	Y (%)	Experimental Conditions	Ref.
1	NbCl_5_	Ionic liquid	0.5	80	95	83	79	0.2 mmol catalyst; 1 mmol fructose in 10 mmol 1-butyl-3-methylimidozolium chloride, [bmim]Cl.	[154]
2	Nb_2_O_5_	Water	6	130	82	40	33	0.08 g catalyst; 0.60 g fructose; 60 mL water.	[155]
3	Sulfated Nb_2_O_5_	DMSO	0.17	120	96	75	72	0.018 g catalyst; 0.18 g fructose; 2 mL DMSO. After 5 h reaction the HMF yield reached 88%.	[156]
4	Nb_2_O_5_	DMSO	2	120	100	86	86	0.01 g catalyst; 0.56 mmol fructose; 5 g DMSO.	[157]
5	Nb_2_O_5_·H_2_O	Water/ 2-butanol	2	140	100	42	42	0.2 g catalyst; 2 mL of 1 wt% fructose; 5 mL 2-butanol.	[140]
6	Nb_2_O_5_·H_2_O treated with H_3_PO_4_	Water/ 2-butanol	0.83	160	90	99	89	0.10 g catalyst; 1.20 g fructose; 20 mL water and 30 mL 2-butanol; 800 rpm.	[139]
7	Niobium phosphate	Water	0.17	180	87	39	34	Fructose/catalyst mass ratio = 10; microwave was used.	[158]
8	Niobium phosphate	Water	0.5	130	58	78	45	0.8 g catalyst; 0.8 g fructose in 10 mL water.	[159]
9	Niobium phosphate	Water/acetone	0.91	138	78	64	50	20 g L^−1^ fructose; fructose/catalyst ratio = 1; acetone/water mass ratio = 1.	[163]
10	Niobium phosphate	Water/DMSO	5	140	95	95	90	10 wt.% catalyst; 1.0 wt% fructose; DMSO/water mass ratio = 1.5.	[159]
11	HNb_3_O_8_	Water	0.3	155	85	66	56	0.02 g catalyst; 1 g fructose; deionized water = 9 g. Microwave irradiation was used.	[164]
12	Nb_2_O_5_/Nb_3_O_7_(OH) modified with H_2_O_2_	DMSO	2	130	100	47	47	0.1 g catalyst; 10 mL of 20 g L^−1^ fructose.	[160]
13	Nb_0.2_-WO_3_	Water	2	120	100	30	30	0.1 g catalyst; 0.01 g fructose; 1 mL H_2_O.	[140]
14	NbPW (Nb/P molar ratio = 0.6)	DMSO	3	80	100	97	97	0.05 g catalyst; 56 mM fructose; 5 mL DMSO.	[161]
15	Nb-NTMPA	DMA	1.5	100	100	86	86	0.05 g catalyst; 0.1 g fructose; 2 g DMA, 0.2 g NaBr.	[165]
16	Nb-doped TiO_2_	Water	0.33	150	97	58	56	12.5 mg mL^−1^ catalyst; 0.55 mmol fructose; 15 W microwave irradiation.	[163]
17	Carbon/Nb_2_O_5_	DMSO	2	120	100	77	77	0.01 g catalyst; 0.1 g fructose; 5 g DMSO, 500 rpm.	[162]

NTMPA = nitrilotris(methylenephosphonic acid). DMA = N,N-dimethylacetamide.

### 3.3. Xylose Conversion to Furfural

Xylose is the most abundant pentose in hemicellulose and is found in a variety of biomass such as corn husks, corn cobs and hardwoods. Xylose can be isomerized into xylulose and then dehydrated in the presence of an acid catalyst in different organic or aqueous phases to produce furfural (Figure 19), which is one of the most important platform molecule in lignocellulosic biomass [166]. In this process, xylose isomerization is catalyzed by Lewis acids, while dehydration is carried out at BrØnsted acid sites [167]. Niobium oxide has been the most often reported niobium catalyst for converting xylose to furfural [138,139,140,141,142,143,144,145,146,147,148,149,150,151,152,153,154,155,156,157,158,159,160,161,162,163,164,165,166,167,168,169,170,171,172,173,174,175,176,177,178]. In addition to Nb_2_O_5_, niobium phosphate and MCM-41 and SBA containing niobium have also been reported [159,175,179]. The most often used solvent is a water/toluene mixture, but other solvent mixtures such as water/THF, DMSO and water/GVL have been used as well. The reaction temperature varies between 120–170 °C, and xylose conversion above 90% can be achieved with selectivity for furfural reaching up to 93%. These data are summarized in Table 6.

With a few additional steps, it is possible to transform furfural into added-value compounds. A proof of this was the synthesis of butyrolactone, which is used in the preparation of pyrrolidones that are applied in the pharmaceutical industry. At first, furfural is converted into 5-hydroxy-2(5H)-furanone (HFO) by photocatalytic oxidation followed by hydrogenation and dehydration over a bifunctional Nb/Zr oxides containing Pt o produce γ-butyrolactone [180].

Another study evaluated the insertion of W6+ and Ti4+ in niobia loaded with Pt particles for the catalytic hydrogenation of furfural to furfuryl alcohol. The presence of Ti4+ in the matrix of Nb_2_O_5_ substantially reduced the acidity of the support, minimizing parallel reactions with furfural and consequently favoring the selectivity toward furfuryl alcohol [167]. Nb_2_O_5_ is a great catalyst in oxidative reactions, but it is not the only one of the niobium pentoxides that has an excellent activity in the oxidation of organic substrates. Poskonin [181] has obtained a 60% yield in the conversion of furfuraldehyde to 2(5H)-furanone after 80 h, applying Nb(OAc)_2_ as a catalyst and hydrogen peroxide as an oxidant in aqueous medium. This value remains impressive even over such a long time, because normally, this type of reaction is quite challenging, requiring two-phase systems with chlorinated solvents and times as long as the above-mentioned to obtain lower yields than that reported by Poskonin.

**Table 6 molecules-28-01527-t006:** Performance of niobium-based catalysts for conversion of xylose into furfural—from 2010 to date.

Entry	Catalyst	Solvent	t (h)	T (°C)	C (%)	S (%)	Y (%)	Experimental Conditions	Ref.
1	Amorphous Nb_2_O_5_	water/toluene	3	120	93	48	45	0.10 g catalyst; 0.075 g xylose; 2 mL water + 3 mL toluene.	[174]
2	Nb_2_O_5_	water/toluene	1.5	170	90	56	50	0.05 g catalyst; 0.15 g xylose; 1.5 mL water + 3.5 mL toluene. 600 rpm.	[175]
3	Nb_2_O_5_	water/toluene	3	120	97	81	79	0.10 g catalyst; 0.075 g xylose; 2 mL water + 3 mL toluene.	[167]
4	Nb_2_O_5_	water/THF	2	130	95	47	45	0.04 g catalyst; 0.12 g xylose; THF/water ratio = 4.	[176]
5	Nb_2_O_5_ nanowires	DMSO	2	120	90	82	74	0.01 g catalyst; 0.1 g xylose; 20 mL DMSO	[177]
6	Niobic acid/niobium phosphate	water	0.5	160	44	75	33	1 g catalyst; 2 wt% xylose	[163]
7	Nb-SBA-15	water/toluene	24	160	85	93	79	4 g catalyst; 20 g L^−1^ xylose; water/toluene volume ratio = 1.	[182]
8	MCM-41-supported niobium oxide	water/toluene	1.67	190	83	55	46	0.05 g catalyst; 0.15 g xylose; 1.5 mL water + 3.5 mL toluene; 1000 rpm. NaCl increases the furfural yield to 60% at 170 °C after 3 h reaction.	[179]
9	H_3_PO_4_ treated Niobium phosphate	water/toluene	1	160	52	43	23	0.14 g catalyst; 2.0 g xylose; 18 mL water + 30 mL toluene.	[183]
10	Nb_2_O_5_/TiO_2_	H_2_O/GVL	-	130	98	30	29	0.02 M xylose in H_2_O/GVL = 1:9 *v/v*, 0.1–0.4 mL min^−1^, residence time = 106 s.	[178]

### 3.4. Conversion of Glycerol to Different Products

The overproduction of glycerol as a by-product in the hydrolysis of fat or plant oils to fatty acid [184], in the saponification of triglycerides to soap [185,186] and in the transesterification of fats or plant oils to biodiesel [186,187,188] requires an intensification of research on the valorization of glycerol toward value-added products. The increasing biodiesel production as an environmentally friendly fuel draws a perspective on the increasing production of glycerol and growing interest in its utilization. Biodiesel as a fuel exhibits a relatively low greenhouse effect because of the lower emission of pollutant gases and particulate matter. Moreover, the use of biofuels eliminates sulfur emissions and the formation of polycyclic aromatic hydrocarbons; besides, they are biodegradable [189,190]. Therefore, in the last decade, the focus has been on the development of catalysts active in different routes of glycerol valorization to valuable chemicals. Three hydroxyl groups in the glycerol molecule make this alcohol highly soluble in polar solvents (such as water and alcohols) and provides glycerol molecules with a high versatility in possible chemical reactions, allowing direct applications facilitating the synthesis of high value-added chemicals. The high glycerol reactivity allows its dehydration, oxidation, total or partial hydrogenation, esterification and etherification, chlorination, ammoxidation, hydrogenolysis or pyrolysis. Niobium-containing catalysts have been already studied for most of these routes of glycerol valorization, as illustrated in Figure 20 and summarized in Table 7. All of them will be discussed below.

#### 3.4.1. Glycerol Dehydration

Catalytic glycerol dehydration can be conducted in either the gas phase [191,192,193] or liquid phase [193,194]. The dehydration of glycerol requires the presence of acidic sites on the catalyst surface. Brønsted acid sites (BAS) are mostly active in a few steps in the dehydration of glycerol to acrolein, whereas Lewis acid sites (LAS) are active in the pathway leading to acetol formation, as illustrated in Figure 21 [191]. Niobium(V) oxide and niobium-containing compounds exhibit both BAS and LAS active centers. The role of Lewis and Brønsted acid sites in the dehydration of glycerol on niobium oxide and Na^+^-exchanged niobium oxide was investigated using FTIR spectroscopy supported by DFT calculations [195]. Two pathways for the dehydration of glycerol have been discussed (Figure 21). The first pathway proceeds via the dehydration of a terminal OH group to form an intermediate enol (2-propene-1,2-diol), which easily tautomerizes to the thermodynamically favored hydroxyacetone. When glycerol interacts with a Brønsted acid site without steric constraints, the dehydration of the secondary alcohol group is strongly preferred due to the higher stability of the corresponding carbenium ion that is formed as the transition state. This reaction leads to the formation of 1,3-propenediol, which tautomerizes to 3-hydroxypropionaldehyde, which is an unstable intermediate that undergoes a second acid-catalyzed or thermal dehydration step to yield acrolein. Such a reaction pathway requires not only the presence of BAS but also a proper porous constraints, which was achieved on niobia studied in the above-mentioned publication. Considering the role of LAS and BAS in glycerol dehydration, the authors compared the IR spectra of glycerol adsorbed at room temperature on niobia having both LAS and BAS and on the other metal oxides: Al_2_O_3_, ZrO_2_ [196] which have only LAS on their surface. The latter two oxides that have only LAS centers did not adsorb glycerol at room temperature, which is in contrast to Nb_2_O_5_ having both LAS and BAS centers. It has been found that the dehydration of glycerol adsorbed on LAS of niobia involved the participation of nearby BAS. In [195] it has been proved that during the adsorption of glycerol on niobia’s LAS, one of the primary OH groups of alcohol dissociates to form bridging alkoxy groups, whereas the second primary OH group is coordinatively but non-dissociatively bonded to metal LAS. The secondary OH group of glycerol can be hydrogen bonded to the basic oxygen atom on the niobia surface. The formed glycerol adsorbed complex is stable and does not easily dehydrate to 2-propene-1,2-diol. If BAS are located in the neighborhood of Lewis acid sites, one of the primary hydroxyl groups can be easily dehydrated. However, when glycerol interacts directly with BAS, the dehydration of the secondary OH groups is strongly preferred due to the higher stability of the carbenium ion that is formed as the transition state leading, via the next dehydration step, to acrolein formation.

The details concerning the dehydration of glycerol on niobium-containing catalysts are summarized in Table 7 entries 1–19. In niobium-containing catalysts, BAS coexist with LAS, and to achieve satisfactory glycerol dehydration selectivity to acrolein, a high BAS/LAS ratio on the catalyst surface is recommended. However, strong BAS lead to coke formation and thus to deactivation of the catalyst. One also has to take into account that in the presence of steam at the reaction temperature, some part of LAS may undergo hydration, forming new BAS, as mentioned in [196]. Thus, the starting BAS/LAS ratio on the catalyst surface can be changed during the reaction. Moreover, not only is the BAS/LAS ratio on the catalyst surface a factor responsible for the dehydration of glycerol to acrolein, textural properties of the catalyst are also very important [197]. The detailed study of niobium-zirconia catalysts in [198] (Table 7 entry 3) allowed drawing a conclusion that the weak and very weak BAS are active in glycerol dehydration. The weak acid sites were about ten times more active than the very weak ones. The variation of selectivity to acrolein on the same catalysts was more difficult to explain and did not correlate well with the acid strength distribution of the active sites, although the catalysts with fewer very weak sites were more selective to acrolein. Thus, it has been proposed that also other parameters such as site distribution, the presence of basic sites, or particles and pore sizes could also play a role.

Both a proper BAS/LAS ratio and the reduced acidity strength of BAS can be achieved by the proper composition of the surrounding of niobium species. Moreover, textural properties of supports for niobium species have to be taken into account in the catalysts addressed to glycerol dehydration. Therefore, niobium species loaded on different supports such as silica ZrO_2_ [197,198], SBA-15 [199], ZrO_2_-doped silica [197,200,201] or Zr-doped silica treated with H_3_PO_4_ [200] were considered by different authors. Niobium oxide in combination with other metal oxides has been also used as a catalyst. Niobium-tungsten mixed oxides were applied in both forms, as a separate mixture of oxides [202,203], or supported on mesoporous silica [203], alumina [204] and zirconia [205]. Niobium–zirconium mixed oxides have been used as well [206]. Phosphorous species as a dopant of niobium compounds appeared to be attractive for control of the catalysts acidity and, therefore, were applied for the construction of catalysts dedicated to glycerol dehydration. Different compositions of such materials have been studied: NbOPO_4_ [207], siliconiobium phosphate [208] and phosphate species loaded on niobia [209] or on W,Nb-mixed oxides [210]. Niobium(V) oxide has been also used as a support for more complex active components such as heteropolyacids (H_3_PW_12_O_40_) [211] or H_3_PW_12_O_40_ doped with cesium and both cesium and phosphorous [212]. Different compositions of the above-mentioned catalysts resulted in different surface properties and different activity/selectivity in glycerol dehydration. According to the data summarized in Table 7, the dehydration of glycerol was performed mainly in the gas phase (with the exception of [200]) in the temperature range between 235 and 350 °C. Glycerol conversion ranged between 22 % for Nb_2_O_5_/SBA-15 [199] and 100% for, e.g., H_3_PW_12_O_40_/Nb_2_O_5_ with 89% yield of acrolein [211].

The application of silica as an unactive support for niobia has proved ineffective as it did not provide a catalyst stable in glycerol dehydration to acrolein because of coke formation on strong BAS [198], as shown in Table 7 entry 1. The best catalyst studied by the above authors, i.e., silica supported with 20% of niobia provided 100% conversion of glycerol (with selectivity to acrolein of 65%) after 2 h of the reaction, whereas after 10 h time on stream, the glycerol conversion decreased to 63% and selectivity to acrolein decreased to 52%. Stawicka et al. [213] (Table 7 entry 2) have found that the silylation of niobium-containing mesoporous silica (Nb-25/SBA-15, the material containing 25 wt% of niobium located in both the skeleton of SBA-15 and in the extra-framework position as niobia) significantly decreased coke formation by lowering of the number of strong BAS. Modification of the niobia acidity can be also achieved by the use of active support such as zirconia [197] (Table 7 entry 3) or zirconia-doped silica [201] (Table 7 entry 4) also treated with H_3_PO_4_ [200] (Table 7 entry 5). The level of coverage of zirconia with niobium species has been identified as an important factor. The most efficient catalysts were those with a better coverage of zirconia support with niobium oxide and doped with niobium [197]. The key issue for the stability of the catalysts was the neutralization of Lewis acid sites of uncovered zirconia, which are unselective coke initiator sites [197]. The stability of the catalysts was related to the formation of cyclic molecules formed by the reaction of acrolein with the products of decomposition of hydroxyacetone, glycerol or acrolein. Niobium supported on zirconia-doped silica [197] allowed a control of the BAS/LAS ratio, depending on niobium loading. This possibility permitted selection of the catalyst (Nb8 wt%) with the highest BAS/LAS ratio and acrolein selectivity but the lowest glycerol conversion. In the following studies [200], the authors checked the influence of phosphoric acid treatment of 8 wt% Nb_2_O_5_ supported on zirconium-doped mesoporous silica (Si/Zr = 5 molar ratio), varying the Nb/P molar ratio between 0.1 and 1. The acid treatment modified the nature of the species present on the catalyst surface. It has been found that the selectivity to acrolein was improved thanks to the presence of the hydrogenphosphate phase, and the catalyst stability was associated to the existence of acid sites of low and moderate strength.

Not only niobia loaded on different supports can generate a variety of BAS/LAS ratios and acidic strength, which are both important in glycerol dehydration to acrolein, but also the formation of mixed niobium–tungsten oxides can control acid–base properties leading to different levels of effectiveness of acrolein production via the dehydration of glycerol. The mixed W-Nb complex oxides have been applied separately [202,203] or loaded on different supports, such as mesoporous silica—KIT-6 [203], alumina [204] and zirconia [202,203]. Nb-substituted tungsten bronzes constitute one of the most selective acid catalysts for the gas phase dehydration of glycerol into acrolein reported in the literature [202,203]. The mixed W-Nb-O oxides in [202] (Table 7 entry 6) were prepared in different manners. Hydrothermal synthesis led to the formation of layered Nb_8_W_9_O_47_ ordered phase, if calcined at 1000 °C (denoted as W-Nb-O (1273 K)), whereas the solid-state method gave Cs_0.5_[Nb_2.5_W_2.5_O_14_] (denoted as Cs-W-Nb-O) with an orthorhombic structure in the crystal structure. Both materials had the same number of acidic sites: a layered arrangement of octahedral units along the c-direction but a different a–b plane structure. They showed different activity and selectivity to acrolein. W–Nb–O (1273 K) revealed an appreciable trend to give higher yields of dehydration products such as hydroxyacetone and lower yields of acrolein when compared to those of Cs–W–Nb–O. This result indicated that the structure of catalysts affected the product selectivity and confirmed the conclusions following from the studies of non-Nb-containing catalysts. In the other paper [214,215] (Table 7 entry 7), the authors documented that increasing the niobium content in hexagonal tungsten bronzes initially favors a higher selectivity to acrolein, but at high Nb contents, heavy compounds and carbon oxides are formed in a large amount because of a higher concentration of LAS. Thus, the selectivity to acrolein decreased when the Nb content in the catalysts increased.

Salvia et al. [204] (Table 7 entry 8) have compared the effectiveness of W-Nb-O mixed oxides used separately (denoted as Nb-W-r) and loaded on an inert support, mesoporous silica with ordered structure—KIT-6 (denoted as NbW/KIT6). The supported catalyst displayed not only the lower density of acid sites but also a higher content of weak acidic sites. Thus, the acid strength of Nb-W-r decreases when it is supported on mesoporous KIT-6. It results in a slightly higher acrolein production over NbW/KIT6, which is principally due to the lower number of side reactions as a consequence of an enhanced desorption of the product. An even greater increase in acrolein yield (83%) was achieved on WNb-H, i.e., the non-supported mixed oxide prepared via the hydrothermal method (in contrast to NbW-r synthesized by the reflux method). It was due to the lowest acid strength of this material. Niobium and tungstate phases were also loaded on an active support i.e., alumina [205] (Table 7 entry 9). Under anaerobic conditions, such catalyst with equimolar amounts of niobia and tungsten oxides (0.5Nb0.5WAl) exhibited very high selectivity to acrolein of above ~74% at full conversion of glycerol. However, the catalyst was deactivated with the time on stream, and it was found that the changes in textural properties including the decreasing pore size caused the deposition of heavy compounds and coke molecules on the pore walls (site blocking). Interestingly, the catalyst was more stable when co-feeding oxygen with glycerol. It was observed that an oxidative atmosphere produced more COx and reduced the formation of heavy compounds. It is worth pointing out that the selectivity to acrolein over 0.5Nb0.5WAl was slightly lower, 68%, under aerobic conditions. Similarly, mixed tungstate–niobium oxides loaded on monoclinic zirconia [205] (Table 7 entry 10) showed 100% conversion of glycerol and selectivity to acrolein of above 70%, although the addition of oxygen to the feed had almost no effect on the yield to acrolein but reduced the deactivation rate. Tungstated zirconia catalysts with and without niobium doping was also applied for glycerol dehydration and studied in [214] (Table 7 entry11). The authors analyzed not only the role of acidic sites but also basic ones. Doping a small amount of niobium to tungstate/zirconia catalyst increased the stability of the catalysts due to a decrease in the number of basic sites. It was found that the niobium dopant was mostly engaged in electronic neutralization of the surface acidity instead of geometric blocking of the basic sites. Moreover, fine tuning of the ratio of acid/base sites was found to be important in achieving good catalytic performance in the gas-phase dehydration of glycerol to acrolein. It was discovered that the ratio of acidic/basic sites in the range of 4–5.4 gave the highest glycerol conversion and acrolein yield and the lowest catalyst deactivation rate. The spent catalysts were easily regenerated by calcination at 450 °C.

The control of acidity in niobium-containing catalysts, important for glycerol dehydration to acrolein, can be also achieved by the addition of a dopant with phosphorous species. The acidic sites distribution on the surface of niobiumoxphosphate material, depending on the calcination temperature, was crucial for reaching a high selectivity to acrolein [207] (Table 7 entry 13). The highest activity during glycerol dehydration and acrolein selectivity was exhibited by niobiumoxphosphate calcined at 550 °C (NbP-550) catalyst and was attributed to the presence of only moderate acidic sites whose majority were Brønsted acidic sites. In [208] (Table 7 entry 14), the authors synthesized siliconiobium phosphate (NbPSi-0.5) that contained a similar amount of acid sites but weaker acidic strength than the highly acidic niobium phosphate. Thanks to these properties, NbPSi-0.5 exhibited a high activity in glycerol dehydration to acrolein, and its stability was three times higher than that of the catalysts without silicon. Such enhancement in catalytic effectiveness was attributed to different behaviors: nearly pure Brønsted acidity, which suppressed side reactions leading to coke formation; a considerable reduction in pore blocking due to the presence of mesopores and a decrease in the amount of coke and oxidation temperature. Phosphate species loaded on niobia [209] (used for liquid phase glycerol dehydration to acrolein—Table 7 entry 15) or on W,Nb-mixed oxides [212] (Table 7 entry 16) also modified the acidity of niobium-containing catalysts. Niobium(V) oxide was likewise applied as a support for heteropolyacid (H_3_PW_12_O_40_) [211] (Table 7 entry 17) or the same heteropolyacid modified with cesium and doped with phosphorous (CsPW) [212] (Table 7 entry 18). The catalytic properties of these catalysts during glycerol dehydration were related to the surface acidic functionality of the catalyst. The loading of CsPW on Nb_2_O_5_ increased the Brønsted acidity and enhanced the acrolein dehydration [208]. It was found that the acrolein selectivity was affected by the reaction temperature, oxygen co-feeding, and glycerol concentration. The highest acrolein selectivity (76.5%) was obtained at 320 °C, O_2_/N_2_/glycerol molar ratio of 1/5/0.16 (mol/mol), and glycerol concentration of 0.2 g/g.

To sum up, niobium species as components of the catalysts dedicated to glycerol dehydration to acrolein can act as (i) active components (BAS coming from, e.g., niobia whose strength can be controlled by additional components, e.g., aminoorganosilane, or by mixing with another metal oxide, or the formation of siliconiobium phosphate); (ii) promoters which reduce disadvantageous activity of other active components (e.g., by electronic neutralization of surface acidity in niobium-tungstate/zirconia catalysts and control of the optimum acid/base ratio); (iii) porous supports, e.g., for heteropolyacid which can provide not only the convenient porosity for the active phase but also control the concentration of BAS via interaction of niobia Lewis acid sites with steam leading to BAS formation and reducing LAS number, thus favoring the acrolein route in glycerol dehydration. As far as the stability increase in the niobium-containing catalysts is concerned, it can be achieved in different ways illustrated in Figure 22.

#### 3.4.2. Glycerol Oxidative Dehydration

The selective dehydration−oxidation (also known as oxidehydration or oxidative dehydration) of glycerol to acrylic acid is a very attractive approach to glycerol utilization. In the acrylic acid market, new processes are searched for that would be able to offer viable alternatives to propylene-based production. Therefore, acrylic acid synthesis from glycerol could be an effective solution addressed to both issues. The preferred realization of such a process would be in one pot. It can be carried out in a two-bed catalytic system (the coupling of two catalysts in one reactor; the upper layer containing the acid catalyst and the bottom layer containing the redox one) or a single-bed system (multifunctional catalyst containing both acidic and redox centers). Irrespective of the system used in the first step of such a reaction, acrolein is formed in the presence of acidic centers on the catalyst surface, and then, redox centers are involved in the oxidation of acrolein to acrylic acid as shown in a simple scheme presented in Figure 23.

A more complex picture of different reaction pathways in the oxidative dehydration of glycerol is shown in Figure 24. The dehydration of glycerol on LAS or BAS present on the catalyst surface is the step determining the kind of products which would be formed (acrylic acid vs. acidic acid). Thus, for the production of acrylic acid, the acrolein route should be followed. Therefore, the catalysts presented in the previous section used in the dehydration of glycerol to acrolein could be considered as a part of the catalyst bed (in a two-bed system) or a base for doping with redox active sites (in a single-bed system) addressed to the oxidative dehydration of glycerol to acrylic acid. All niobium-containing catalysts used in glycerol oxidative dehydration contained vanadium species (Table 7 entries 20–23) necessary for the acrolein oxidation step. The main catalysts containing vanadium and niobium and studied in a one-pot oxidative dehydration of glycerol to acrylic acid include hexagonal tungsten bronzes (HTBs) [210,215,216,217,218,219] with in-framework or extra-framework vanadium species. Remarkable structure–reactivity correlations have been revealed in [135]. It has been evidenced that the majority of vanadium in VO-WOx is incorporated as extra-framework octahedral species. Such species are not sufficiently stable. Only if another transition metal element, such as niobium, enters the hexagonal tungsten oxide framework does the HTB structure become stable. Chieregato et al. [219,220] have defined the most important features required from the effective multifunctional catalyst applied in this reaction. Such a catalyst should contain BAS of medium acid strength for the selective dehydration of glycerol to acrolein, octahedral VO_6_ species located in the tungsten bronze which would activate the selective oxidation of acrolein to acrylic acid. An important task is to eliminate strong acid sites, which could be involved in the consecutive reactions of acrolein and acrylic acid as well as avoid too close proximity of acid sites which would impede the desorption of acrylic acid. The incorporation of Nb^5+^ into W-V-O bronze decreases the density of acid sites and increases the fraction of stronger acid centers of medium strength optimal for this reaction. In [219], the optimum composition of the tri-component system has been established. The best results were obtained for the catalyst containing V/(W + V + Nb) = 0.13 and Nb/(W + V + Nb) = 0.13 (atomic ratios) which gave 34% yield to acrylic acid (with overall selectivity to acrylic acid and acrolein of 51%) at 290 °C.

It has been also evidenced [219] that the selectivity to acrolein and acrylic acid was strongly affected by the reaction conditions—temperatures higher than 300 °C or contact time higher than 0.15 s favored the formation of CO and heavy compounds. Tungsten bronzes (W-V-O and W-Nb-O) supported on mesoporous silica (KIT) as an inert support have been studied in [199]. However, the catalyst containing only tungsten and niobium was not active in glycerol oxidehydration reaction but only in the dehydration process. For W-V-O bronze supported on KIT, a decrease in both the number of the strong acid sites and the degradation of acrolein and acrylic acid were obtained. Such an effect could be achieved thanks to a strong interaction of small crystals of V-containing tungsten bronze with the support.

As far as the reaction conditions are concerned, Chieregato et al. [220] have investigated the role of oxygen content in the gas stream. Oxygen plays the fundamental role of accelerating the oxidation of the intermediately formed acrolein into acrylic acid by allowing a greater concentration of the oxidizing V^5+^ sites. The optimal control of the two consecutive steps of the oxidative dehydration of glycerol to acrylic acid, and of the parallel reaction of acrolein transformation into by-products (ketals and oligomers), was achieved in the presence of a defined glycerol-to-oxygen inlet ratio shown in Table 7 entry 21. Under these reaction conditions, a maximum acrylic yield of acrylic acid of 50.5% was achieved. It was accompanied by an increase in productivity by more than one order of magnitude because of both the greater concentration of glycerol used in the inlet feed and shorter contact time needed.

Omata et al. [188,209] have also prepared W–V–Nb–O complex metal oxides of various compositions of the structure similar to that of orthorhombic Mo_3_VO_x_, so they were different than that described by Chiereagto et al. [219,220]. Such materials were found to be efficient catalysts working for the gas-phase direct oxidative transformation of glycerol to acrylic acid [Table 7 entry 22]. Similarly as in [219,220], BrØnsted acidity important for glycerol dehydration to acrolein was improved by a combination of Nb and W mixed oxides, whereas doping with vanadium prominently promoted the formation of acrylic acid in the glycerol transformation in the presence of oxygen but showed no effect on the dehydration of glycerol to acrolein. The optimum elemental composition for the reaction was W_2.2_V_0.4_Nb_2.4_O_14_ giving ca. 46% yield of acrylic acid under W/F = 6.7 × 10^−3^ gcat min L^−1^ and 37% under W/F = 1.0 × 10^−2^ gcat min L^−1^. The addition of phosphoric acid to this catalyst allowed an increase in acrylic acid yield to 59% under W/F = 1.0 × 10^−2^ gcat min L^−1^. Such a yield was almost constant for 5 h of the reaction. Phosphoric acid not only caused an increase in the amount of BAS but also interacted with V sites for suppressing the sequential oxidation of acrylic acid to CO_x_.

The coupling of two catalysts (dehydrating and oxidizing) as two in-series fixed beds in a single reactor is generally accepted to lead to higher yields of acrylic acid [136,138]. However, the two beds have to be combined in such a manner that both acidic catalyst and redox one would show optimum performance at the same temperature and oxygen to glycerol ratio. As shown in [146], the use of Nb/SBA-15 as acidic catalyst and V/SBA-15 as redox catalyst did not lead to an oxidehydration process toward the acrylic acid but only to dehydration to acrolein and overoxidation to CO_x_. A two-bed system has been successfully applied in [215] in which Cs_2.5_H_0.5_PW_12_O_40_ supported on Nb_2_O_5_ (CsPW-Nb) was used as a dehydration catalyst (Table 7 entry 23), whereas the oxidation catalyst was made of vanadium−molybdenum mixed oxides supported on silicon carbide (VMo−SiC). The experimental results showed that the optimum reaction temperature and oxygen ratio for these two catalysts were very similar. It was very important because serious limitations can derive from being forced to use the same reaction conditions for both catalytic beds. The authors indicated that compared with the single-bed system, the two-bed one was more inclined to avoid the overoxidation of glycerol. Interestingly, the presence of byproducts produced on Lewis acid sites in the dehydration step performed on CsPW-Nb and water in the glycerol feed did not show negative effects on the acrolein oxidation reaction. Moreover, the authors have found that the migration of H+ and Cs in CsPW-Nb led to the formation of an almost uniform solid solution and the protons were distributed widely on the surface, which was favorable to the protonation of glycerol and its transformation via the acrolein route of dehydration (Figure 24). Both the CsPW-Nb and VMo-SiC catalysts were stable for at least 70 h, and 75% yield of acrylic acid was achieved.

To sum up, the main role of niobium in the catalysts applied for the oxidative dehydration of glycerol to acrylic acid is its promoting effect. Niobium dopant in vanadium–tungsten bronzes increases the stability of this catalyst used in a single-bed system. Moreover, Nb^5+^ included to the bronze structure modifies the acidity of this catalyst by increasing the acidic sites dispersion and expanding the fraction of medium strength BAS, which are the most favorable in the first step of glycerol oxidehydration. In a two-bed system, niobia acts as a support for cesium-modified heteropolyacids, which optimizes the acidity in CsPW-Nb.

#### 3.4.3. Oxidation of Glycerol to Glyceric Acid

Niobia has been used as a support for gold catalysts applied in the selective oxidation of glycerol to glyceric acid in liquid phase and alkali media [185,186,187] (Table 7 entries 24, 25). It is known that the selective oxidation of glycerol can lead to various valuable oxygenates such as glyceric, tartronic, glycolic, hydroxypyruvic, formic, and oxalic acids. Of course, the crucial research target is to control selectivity for the desired product. The first step of the reaction is the abstraction of a proton from the glycerol molecule with the participation of Lewis acid sites. This process leads to glyceric aldehyde or hydroxyacetone. The next step is aldehyde oxidation to glyceric acid and it is very fast, faster than the oxidation of ketone, and therefore, glyceric acid often dominates among the reaction products (scheme in Figure 25). The glycerol oxidation is a structure-sensitive reaction. It has been indicated for Au-carbon catalysts [216,217,218,219,220,221,222,223] that the size of the gold particle influences the catalytic activity and selectivity. For gold particles sizes between 2 and 47 nm, the smaller the gold particles, the higher the activity in glycerol oxidation. Large Au particles (>20 nm) were more selective to glyceric acid, whereas the smaller ones exhibited a higher selectivity to glycolic acid. Thus, the kind of support for gold plays a crucial role in the control of Au particle size and thus the activity and selectivity of the reaction.

In [185], V_2_O_5_, Nb_2_O_5_ and Ta_2_O_5_ were used as supports for gold, and the activity of these catalysts in the oxidation of glycerol was compared with that of gold loaded on Al_2_O_3_, TiO_2_ and carbons. It has been found that the nature of metal oxides used as supports for gold species determines their acid–base properties, which influences both the particle size of metallic gold loaded on their surface and the final activity and selectivity in the oxidation of glycerol. Gold loaded on alumina, having basic centers, formed larger particles than on niobia, which has acidic ones. From among all catalysts based on metal oxides used, the Au/Nb_2_O_5_ catalysts, prepared by the gold–sol method on crystalline niobia as a support, was attractive for the liquid phase oxidation of glycerol showing 67% conversion of glycerol and 31% yield of glyceric acid at 60 °C and 5 h of the reaction time. Moreover, the ability of reuse of this catalyst was pointed out. Interestingly, when amorphous niobia was applied as a support for gold, both the activity and glyceric acid yield were lower (Table 7 entry 24). The crystallinity of niobia support appeared to be important. Crystalline niobia contains unsaturated niobium atoms at the corners and edges of the crystals, which have a stronger interaction with the gold precursor than saturated Nb species in amorphous niobia. Such an interaction enhances gold dispersion and could increase the activity of oxygen from Nb_2_O_5_, which is active in the abstraction of proton from glycerol in the first step of the reaction. The introduction of copper to the Au-niobia system [186] led to a decrease in the activity and selectivity to glycerol acid. Copper dopant significantly increased the selectivity to glycolic acid because copper species were active in the oxidative dehydrogenation of glyceric acid to tartronic acid, which is next transformed to glycolic acid (via decarboxylation).

The above description of glycerol oxidation on gold catalysts based on niobia clearly shows that niobium(V) oxide not only plays the role of a support for the active gold phase but also is the structural promoter for gold crystallites active in this reaction.

#### 3.4.4. Esterification of Acetic Acid with Glycerol

One of the attractive transformations of glycerol to valuable products is its reaction with acetic acid toward the formation of triacetylglycerol (TAG) (Figure 26). The esterification of acetic acid with glycerol can turn waste glycerol to triacetylglycerol, which is a valuable biodiesel additive that improves the quality of biofuel. In particular, it increases viscosity, cold resistance and anti-knocking properties [225]. Therefore, the acetic acid esterification with glycerol offers two advantages: utilization of waste product and improvement in biodiesel quality and yield (with respect to triglycerides transesterification). As shown in Figure 26, the esterification of acetic acid with glycerol can lead to the formation of all, mono-, di- and triacetylglycerols (MAG, DAG and TAG, respectively). The reaction between acetic acid and glycerol toward acetins formation is catalyzed by acidic catalysts. Many different solids have been used as catalysts, and different operational conditions have been applied in this reaction. The results of these studies indicated that the selectivity to TAG is limited by various features, including the presence of water, shifting equilibrium and weakening the catalyst acid strength, as well as the sequential acetylation of the hydroxyl groups. The porosity of the catalysts can also limit the formation of bulky TAG. Small pores in catalysts favor the production of MAG. Because zeolites (e.g., H-ZSM-5, H-Y, H-mordenite) are known as typical acidic catalysts, they were tested in the first attempts in the reaction between glycerol and acetic acid [223,224,226,227]. However, because of their relatively small pore sizes, they could not be effective in the esterification of bulky molecules. Another group of strong acidic materials, natural and synthetic resins used in this reaction, are not sufficiently stable. Therefore, in the last decade, the focus has been on the modification of ordered mesoporous silicas toward the production of acidic catalysts addressed to the esterification of acetic acid with glycerol. Such materials containing niobium as well as niobium compounds will be discussed with respect to their use in this reaction (Table 7 entries 26–32).

Kim et al. [189] have studied the catalytic performance of several acidic materials in the esterification of acetic acid with glycerol, including dodecamolybdophosphoric acids (HPMo) supported on Nb_2_O_5_ (HPMo/Nb_2_O_5_) and mesoporous SBA-15 (HPMo/SBA-15) as well as PrSO_3_H-SBA-15. As follows from a comparison at the same acid loading and a similar glycerol conversion level (~30%) below the equilibrium, the glycerol conversion turnover rate toward di- and triacetin was considerably higher on sulfonated ordered mesoporous silica, PrSO_3_H-SBA-15, than on HPMo/Nb_2_O_5_ ≥ HPMo/SBA-15. Interestingly, when the performance of similar types of the solid acids were compared, it was found that the acid strength affected the rate and selectivity: the higher the acid strength, the higher the reaction rate. However, no linear relationship between glycerol conversion turnover rate and acid strength was achieved. The authors postulated that the orders of magnitude higher glycerol conversion turnover rates in the case of moderate BrØnsted acid strength on the sulfonic acid functionalized catalysts suggest that the configuration of surface acid moieties contribute substantially to their catalytic activity in the esterification reaction. Moreover, the porosity of the catalyst materials should also be taken into account.

Other authors have reported a series of studies in which niobium containing mesoporous ordered silica (NbSBA-15 and NbMCF) was modified with sulfonic species (formed by oxidation with H_2_O_2_ of MPTMS (MP) = (3-mercaptopropyl)trimethoxysilane anchored to niobiosilicate materials), which were the source of BrØnsted acid sites [190,191]. It has been evidenced [190] that the addition of niobium into the synthesis gel (for the production of material based on SBA-15) enhanced the transformation of thiols in MPTMS to sulfonic species via the oxidation by hydrogen peroxide (MP-NbSBA-15) in comparison with silicate (MP-SBA-15) material prepared in the same way. The generation of sulfonic species was also more effective via the post-synthesis oxidation of silicate sample; however, the sulfonic groups were less stable than in niobiosilicate materials. Thus, niobium, as an additive, enhanced the effectiveness of oxidation of thiol groups in MPTMS and increased the stabilization of sulfate species anchoring on the catalyst surface. Moreover, it was evidenced that the introduction of niobium distorted the long-range ordering of pores because of the interaction of Nb species with MPTMS (co-template in the synthesis of SBA-15), which appeared to be effective in the esterification of acetic acid with glycerol. At a low excess of acetic acid in the reaction mixture (glycerol:acetic acid = 1:3), the presence of niobium in the silica structure (NbSBA-15; Si/Nb = 65) enhanced both the activity (69% vs. 74% of glycerol conversion on MP-SBA-15 and MP-NbSBA-15, respectively) and the selectivity to TAG (19% vs. 28% on MP-SBA-15 and MP-NbSBA-15, respectively). The increase in acetic acid content (glycerol to acidic acid ratio of 1:9) gave rise to almost the same TAG selectivity (35% vs. 36% on MP-SBA-15 and MP-NbSBA-15, respectively). The increase in niobium content (Si/Nb ratio of 28) at the glycerol to acid ratio of 1:9 allowed the achievement of a higher TAG selectivity (41%) with 73% of glycerol conversion (it led to 30% of TAG yield). A higher glycerol conversion (89%) was obtained if mesostructured cellular foam (MCF) consisting of uniform spherical cells ca. 20–40 nm in diameter modified with niobium (Si/Nb = 129) and oxidized MPTMS (MP-NbMCF) was used as a catalyst with a glycerol to acidic acid ratio of 1:9 [228]. Thus, the effect of the catalyst’s porosity was evidenced. An increase in the activity in esterification of acetic acid with glycerol, under the same reaction conditions, was reached if MCF was additionally modified with phosphorus (MP-NbPMCF) [228]. In the presence of this catalyst, the highest yield of TAG (36%) along with 99% of glycerol conversion was obtained. The presence of niobium in the MP-NbPMCF catalyst enhanced the efficiency of thiol species oxidation, increasing the BAS number, whereas the phosphorus species generated additional BAS. The increase in the number of BAS from these two sources led to the enhancement of activity and TAG formation in the esterification of acetic acid with glycerol.

To improve the effectiveness of TAG production from glycerol in the presence of niobium-containing catalysts, acetic acid has been replaced by acetic anhydride [191]. From among the list of catalysts studied by the authors, niobium phosphate was applied in both the reaction of glycerol with acetic acid and with acetic anhydride at 120 °C for 80 min with glycerol: an acetic acid/acetic anhydride ratio of 1:4. In both cases, 100% conversion of glycerol was reached, whereas the selectivity to TAG was only 7% if acetic acid was used as a reagent and 100% for the reaction with acetic anhydride.

Summarizing, niobium addition to ordered mesoporous silicas (SBA-15 and MCF) allows an increase in the efficiency of oxidation of thiol species (in MPTMS anchored on niobiosilicate materials) and in this way increases the BAS number. Moreover, niobium addition disturbs the long-range ordering of pores as a result of the interaction of Nb species with MPTMS. This phenomenon appeared to be an advantage in the esterification of acetic acid with glycerol.

#### 3.4.5. Acetalization of Glycerol

The acetalization of glycerol with acetone is an important reaction that allows the transformation of glycerol to fuel additives, such as solketal, which is characterized by lower polarity and viscosity, and therefore, greater volatility; thus, it is able to improve the combustion efficiency [229,230]. Solketal is applied not only as a fuel additive, but also, it is important organic compound that may act as a solvent, plasticizer, surfactant, flavor enhancer, and pharmaceutical intermediate.

The catalytic reaction between glycerol and acetone requires the presence of acidic centers on the catalyst surface and produces two structural isomers, 5-membered ring 2,2-dimethyl-1,3-dioxolane-4-yl methanol (solketal) and 6-membered ring 2,2-dimethyl-1,3-dioxan-5-ol, as well as water as a by-product [230] (Figure 27). However, this reaction affords mainly one of these isomers, the five-membered ring—solketal [231,232,233,234]. The theoretical calculations [230] suggested that the five-membered ring solketal is thermodynamically more stable because in six-membered ring 2,2-dimethyl-1,3-dioxan-5-ol, isomer steric repulsions occurs, which is triggered by the presence of a methyl group in the axial position of the six-membered ring. Many articles on glycerol acetalization with acetone in the presence of solid acid catalysts have been published, in which Lewis acidic sites (LAS) [235], BrØnsted acidic sites (BAS) [229,233] or both [219,233,236,237] have been shown to promote the reaction.

Not only the nature and strength (the stronger the acidity, the higher the solketal selectivity [198,238,239]) of acidic centers on the catalyst surface play an important role in the effective production of solketal in the acetalization of glycerol, but the latter also depends on other features. Hydrophobicity of the catalyst is a very important parameter because water formed in this reaction weakens the acidic sites strength and can favor the reverse reaction in which the desired product (solketal) is hydrolyzed back to glycerol and acetone. Thus, the enhancement of the hydrophobic character of the catalyst surface promotes the removal of adsorbed water and in this way preserves the catalyst acidity, hindering the reverse reaction and minimizing the competitive H_2_O adsorption [229,233,237]. Another parameter which has to be considered is the catalyst porosity—the larger the pores, the easier the diffusion of reactants and the higher the effectiveness of the reaction. In addition to the limiting effect of the catalysts properties on the productivity of solketal in the acetalization of glycerol, the reaction conditions have to be carefully controlled, mainly the glycerol:acetone ratio [195,196,236,240], pH [195,196,233], and the source of glycerol [195,233].

Niobium-containing catalysts applied in the acetalization of glycerol described in the literature and listed in Table 7 (entries 34–41) can be divided into three groups: (i) the catalysts having BAS and LAS (predominantly BAS) originated from niobium species—niobium(V) oxides [195,196,198] and mixed oxides (having predominantly BAS) [233]; (ii) those showing exclusively Lewis acidity coming from niobium included into the skeleton of mesoporous silica [233,237]; (iii) those based on molecular sieves containing BAS in which niobium species played the role of a promoter [195,197]. One of the first literature reports on the use of niobium(V) oxide as a catalyst in the acetalization of glycerol appeared in [198]. The authors used niobia calcined at different temperatures from the range of 200–700 °C. With increasing calcination temperature, the crystallization of niobia grew deeper and the surface area as well as acidity of the materials decreased. The optimum activity (80% conversion of glycerol) and selectivity to solketal (92%) were achieved at 70 °C after 6 h of the reaction in the presence of amorphous niobia (calcined at 300 °C). It has been documented that the acid strength of niobia depended on the calcination temperature and higher acid strength favored higher catalytic performance. Moreover, the catalyst could be reused without deactivation. However, the authors did not find a direct correlation between the nature of acidic sites (BAS/LAS) and the activity and selectivity of the reaction. Such a correlation could be found if the catalyst included only BAS or only LAS as shown in [233], which will be discussed later.

An interesting approach was applied in [195,196] whose authors produced amorphous niobium oxyhydroxide (NbO_2_(OH)) from niobium(V) chloride precursor or also from ammonium niobium oxalate [196], and the synthetized materials were partially hydrophobized by treatment with a surfactant (CTAB, cetyltrimethylammoniumbromide). Moreover, these authors used three different sources of glycerol: commercial grade p.a., desalinated (1% salt) and the crude residue; the latter two originated from biodiesel production by Petrobras S/A. The activity of niobium oxyhydroxide and partial hydrophobized niobium oxyhydroxide in glycerol acetalization was related to that obtained on commercial niobium(V) oxide (Nb_2_O_5_·xH_2_O, from CBMM). It has been proved that the highest activity (65%) after 1 h of the reaction at 70 °C with 95% selectivity to solketal was obtained on partially hydrophobized niobium oxyhydroxide prepared from ammonium niobium oxalate [196]. Under the same reaction conditions, Nair at al. [198] have obtained 30% of glycerol conversion on amorphous niobia (material calcined at 300 °C). It has been proposed in [196] that the best conversion of partially hydrophobized niobium oxyhydroxide might be associated with the presence of surfactant groups because they should facilitate the interactions between the glycerol and acetone substrates, which are immiscible. Such a catalyst is both hydrophobic and hydrophilic, and therefore, it occupies the interface between the liquids. Thus, it is closer to the substrate to facilitate the catalytic process. It is important to point out that the acidity and the degree of hydrophobicity must be balanced to synergize these two properties and obtain the optimum catalyst for glycerol acetalization. The partially hydrophobized niobium oxyhydroxide studied in [195,196] was very stable, and after many reuses, it showed glycerol conversion between 70 and 80%. In the reaction with the optimum ¼ glycerol/acetone molar ratio, the glycerol conversion was still greater than 80% after some reaction cycles [195].

From the practical point of view (a possible commercialization of this process), an interesting study concerned the application of residual glycerol from biodiesel production or desalination in the acetalization reaction [195,196]. When the glycerol coming from biodiesel production was used, the catalysts exhibited much lower activity. It has been suggested that the low conversion of the residual glycerol could be attributed to the presence of water, salt and other impurities formed during biodiesel synthesis that contaminate the residual crude glycerol. Therefore, when desalinated glycerol was used and the reaction was performed on partially hydrophobized niobium oxyhydroxide synthetized from ammonium niobium oxalate, the glycerol conversion was almost the same as that achieved when commercial glycerol p.a. was used [196]. Thus, the authors concluded that the catalysts studied in their work can be efficiently used for converting residual glycerol after desalination to form solketal in an acetalization reaction.

Both Lewis (predominant) and BrØnsted acidic sites (Nb-OH-Al and Nb-OH-Nb species) were identified in niobium-aluminum mixed oxides as active in the acetalization of glycerol to solketal [233]. The catalysts containing a high amount of aluminum oxide (Al/Nb molar ratio of 5 and 10) as well as the solids synthesized in an acidic medium were inactive (glycerol conversion lower than 5%). Therefore, the authors used high-throughput techniques, and the catalytic materials were prepared with the molar ratios Nb:Al = 1:1; 1:0.6; 1:0.3; 1:0.1; and 1:0.05. The final materials were calcined at 350 °C. Depending on the composition, the materials of different structures were obtained. In the samples with 1Nb:1Al and 1Nb:0.6Al compositions, the samples amorphous state dominated, whereas mixed oxides with 1Nb:0.1Al and 1Nb:0.05Al revealed the presence of crystalline AlNbO_4_. The LAS/BAS ratio was not strongly affected by the Nb/Al ratio, but a lower Al content led to a lower concentration of acidic sites. The highest glycerol conversion of 84%, with 98% of selectivity toward solketal, was reached for the sample with the ratio 1Nb:0.05Al at the acetone:glycerol ratio of 4:1. The lower activity of the samples with 1Nb:1Al and 1Nb;0.6Al in spite of a higher concentration of acidic centers was related to the highest hydrophilicity of materials containing a higher content of aluminum. The high hydrophilicity of catalysts favors the reverse reaction of solketal hydration to glycerol.

The above-described catalysts had both BAS and LAS and, therefore, it was impossible to distinguish between the roles of the two types of acidic centers in the acetalization of glycerol. In order to evaluate the effect of particular types of acidic sites, the catalysts based on mesoporous cellular foams (MCFs) containing exclusively LAS—(NbMCF), solely BAS—MP-MCF (MP = (3-mercaptopropyl)trimethoxysilane) and CS-MCF (CS = 2-(4- chlorosulfonylphenyl)ethyltrimethoxysilane as well as both LAS and BAS—MoMCF have been prepared in [236] and studied in glycerol acetalization to solketal by means of Raman monitoring. The application of MCF with large cavities allowed a reduction in diffusion limitation for the formation of acetone–glycerol adduct observed as O···/C=O vibrations band in Raman spectra. Although the adduct formation does not limit the rate of the reaction, its conversion to the product is its rate-limiting step. Thus, the fast formation of a acetone–glycerol adduct leads to a high productivity of the reaction.

The authors of [236] have confirmed that the activity of the catalysts with BAS depends on the number and strength of acidic sites. Glycerol conversion of 87% with almost 100% selectivity to solketal was achieved on the most active BAS type catalyst (CSMCF) with the highest number and strength of acidic sites. The lowest conversion (48%) of glycerol was obtained with NbMCF, i.e., the catalysts containing exclusively LAS. Thus, Brønsted acidic sites are far more efficient than LAS in conversion of the acetone-glycerol adduct into solketal. Moreover, it was indicated that the catalysts containing exclusively BAS were stable if used in the second run after the activation, whereas the catalyst containing solely LAS lost the activity after the first run of the reaction because of the transformation of niobium active phase. The reaction pathways with the participation of BAS and LAS on the catalysts surface have been proposed in [231] and are shown in Figure 28. On the basis of Raman monitoring studies performed in [231,236], the following order of activity of acid catalysts has been proposed, depending on the type of acidic sites: BAS > BAS/LAS > LAS.

The activity of the LAS-containing catalysts in glycerol acetalization can be enhanced by the application of microwave heating of the reactor instead of conventional heating. In [196], the activity of the catalysts based on ordered mesoporous molecular sieves (NbMSU and NbMCM-16) in glycerol acetalization was determined when using two modes of heating: (i) the conventional heating and (ii) microwave radiation heating. The application of microwave radiation led to a higher conversion of glycerol compared to that obtained under conventional heating and to shortening of the reaction time. Moreover, a new reaction product was observed if NbMSU was applied, i.e., under the microwave radiation, the dehydration of glycerol to 1-hydroxy-2-propanone (acetol) was observed.

As mentioned at the beginning of this section, in the third group of niobium containing catalysts applied in glycerol acetalization to solketal, niobium species acted as a promoter and the catalysts had active BAS coming from the other sources [197,231]. In [197], faujasite zeolite-supported niobium-based catalysts (with Nb_2_O_5_ amounts of 5 and 15 wt%) prepared by the impregnation of NaY, HY, and HUSY zeolites (HUSY = protonated form of ultrastable Y zeolite) using niobium(V) ammonium oxalate complex were used. The catalysts had both BAS and LAS in the amounts dependent on niobium loading, but the number of each type of acidic centers was not estimated; only the total acidity was determined. The presence of Nb was proved to increase the hydrophobicity of the zeolites and to affect surface acidity. The mesoporous Nb-HUSY catalysts showed the best catalytic performance (60% of glycerol conversion with 98% selectivity to solketal at 70 °C) and stability within three runs. A comparison of Nb-HUSY activity with that of HUSY (29% of glycerol conversion) clearly indicated that niobium modification enhanced the activity of the zeolite catalyst, although such a modification led to a decrease in the total number of acidic sites.

Another promoting role of niobium was described in [231] whose authors prepared niobium containing MCF (mesoporous niobiosilicate cellular foam) further modified with (3-mercaptopropyl)trimethoxysilane (MP), which was followed by hydrogen peroxide treatment. In this way, they obtained sulfonated catalysts containing BAS. The modification of MCF with niobium prior to MP loading enhanced the number of BAS and thus increased the catalyst activity in glycerol acetalization. The role of niobium species was to increase the efficiency of –SH (from MP) oxidation to sulfonic species (BAS). Interestingly, the results obtained by real-time Raman monitoring confirmed the proposed mechanism of the reaction, which proceeded via the formation of the 3-(2-hydroxypropan-2-yloxy)propane-1,2-diol intermediate, whose presence was confirmed by Raman spectroscopy.

To sum up, in the acetalization of glycerol to solketal, the niobium species in the catalyst composition may act as BAS and LAS (both active in this reaction, but BAS is more efficient) or may act as promoters enhancing the generation of BAS via the oxidation of thiol groups or increase the hydrophobicity of the catalysts. The presented results indicated that both the acidic and the hydrophobic character of the catalyst surface (enhanced by niobium species) as well as the presence of mesoporosity are important for the achievement of a very high catalytic performance in the glycerol acetalization reaction.

**Table 7 molecules-28-01527-t007:** Performance of niobium-based catalysts for conversion of glycerol into different products—from 2010 to date.

Entry	Catalyst	Solvent/Reactant	t (h)	T (°C)	Product	C (%)	S (%)	Y (%)	Experimental Conditions	Ref.
43	Niobium oxyhydroxide	-	6	250	allyl alcohol	91	46	42	Catalyst loading = 2.5 mg/mL; Oxidant = H_2_O_2_ Concentrated glycerol volume = 20 mL.	[184]
24	Au/Nb_2_O_5_ crystalline Au/Nb_2_O_5_ amorphous	Water	5	60	glyceric acid	67 31	47 55	31 17	Catalyst loading = 0.2 g; p_O2_ = 6 bar; 1 M aqueous solution of glycerol; NaOH/glycerol molar ratio = 2.	[185] [187]
25	Au/Nb_2_O_5_	Water	5	90	glyceric acid	93	45	42	Catalyst loading = 0.05 g; p_O2_ = 6 bar; 1 M aqueous solution of glycerol; NaOH/glycerol molar ratio = 2.	[186]
20	W-V-Nb-O	Water		290	acrylic acid	100	34	34	Feed composition: 2 mol% glycerol, 4 mol% oxygen, 40 mol% water, and 54 mol% helium.	[219]
21	W-V-Nb-O	Water	37 ^b^	265	acrylic acid	100	50	50	Feed molar ratio O_2_/Gly/H_2_O/He = 12/6/40/40, residence time 0.15 s.	[220]
22	W_2.2_V_0.5_Nb_2.3_O_14_ W_2.2_V_0.5_Nb_2.3_O_14_ H_3_PO_4_/W_2.2_V_0.5_Nb_2.3_O_14_	Water	5 ^b^	285	acrylic acid	100	46 37 59	46 37 59	W/F = 6.7 × 10^−3^ g_cat_ min L^−1^ W/F = 1.0 × 10^−2^ g_cat_ min L^−1^ W/F = 1.0 × 10^−2^ g_cat_ min L^−1^ Catalyst loading = 0.2 g; Flow rate = 80 mL min^−1^; time on a stream: 1–2 h; water/glycerol molar ratio = 5; composition of reaction gas: Glycerol/O_2_/N_2_/H_2_O = 5/14/56/25 (mol%).	[188] [209]
23	20CsPW-Nb + VMo-SiC	Water	70 ^b^	300	acrylic acid	100	75	75	Catalyst loading (two-bed system) = 0.50 g 20CsPW-Nb, 0.5 g VMo-SiC 40 mL/min gas flow rate (O_2_ = 6 mL/min), 20 wt% glycerol solution fed at 0.6 mL/h (0.24 h^−1^ glycerol WHSV).	[215]
26	HPMo/Nb_2_O_5_	Acetic acid	8	80	monoacetin diacetin triacetin	87	82 17 1	71 15 1	Catalyst loading = 5 wt% based on glycerol feed; Acetic acid/glycerol molar ratio = 6.	[189]
28	MP-NbSBA-15 ^a^ Si/Nb = 65 SH in MPTMS oxidized to SO_3_H with H_2_O_2_	Acetic acid	4	150	monoacetin diacetin triacetin	66	11 53 36	7 35 24	Catalyst loading = 0.1 g; acetic acid/glycerol molar ratio = 9.	[190]
31	Niobium phosphate	Acetic acid	1.3	120	monoacetin diacetin triacetin	100	38 49 7	38 49 7	Catalyst loading = 2 g; Acetic acid/glycerol molar ratio = 4.	[191]
27	MP^a^-NbSBA-15 ^a^ Si/Nb = 28 SH in MPTMS oxidized to SO_3_H with H_2_O_2_	Acetic acid	4	150	monoacetin diacetin triacetin	73	10 49 41	7 36 30	Catalyst loading = 0.10 g; Acetic acid/glycerol molar ratio = 9.	[192]
30	MP/NbPMCF ^a^ SH in MPTMS oxidized to SO_3_H with H_2_O_2_	Acetic acid	4	150	monoacetin diacetin triacetin	99	11 53 36	11 52 36	Catalyst loading = 0.10 g; Acetic acid/glycerol molar ratio = 9.	[193]
29	MP-NbMCF ^a^ Si/Nb = 129 SH in MPTMS oxidized to SO_3_H with H_2_O_2_	Acetic acid	4	150	monoacetin diacetin triacetin	89	11 51 38	10 45 34	Catalyst loading = 0.10 g; Acetic acid/glycerol molar ratio = 9.	[228]
32	Niobium phosphate	Acetic anhydride	1.3	120	triacetin	100	100	100	Catalyst loading = 2 g; Acetic anhydride/glycerol molar ratio = 4.	[191]
33	Sulfonated Nb_2_O_5_	*Tert*-butyl alcohol	5	120	mono-*tert*-butyl-glycerol	95	47	45	Catalyst loading = 5 wt%; TBA/glycerol molar ratio = 4.	[194]
34	Amphiphilic Niobium oxyhydroxide	Acetone	1	70	solketal	73	95	69	Catalyst loading = 0.2 g; Acetone/glycerol molar ratio = 2; TOF = 1106 h^−1^.	[196]
35	Hydrophobized niobium oxyhydroxide	Acetone	1	70	solketal	65	95	62	Catalyst loading = 0.2 g; Acetone/glycerol molar ratio = 4.	[196]
37	1Nb/0.05Al oxides	Acetone	6	50	solketal	84	98	82	Catalyst loading = 2.7 wt%; Acetone/glycerol molar ratio = 4.	[233]
38	Nb/MCF	Acetone	3	40	solketal	48	99	47	2 wt% catalyst, 40 mmol glycerol, 80 mmol acetone Acetone/glycerol molar ratio = 2.	[236]
39	Nb-incorporated SBA-16	Acetone	0.7	30	solketal	86	92	79	Acetalization using microwave; Catalyst loading = 0.05 g 25 mL of glycerol/0.02 M acetone; TON = 2200 molecules/Nb site.	[196]
40	MP-NbMCF ^a^ SH in MPTMS oxidized to SO_3_H with H_2_O_2_	Acetone	3	40	solketal	80	97	78	2 wt% catalyst, 40 mmol glycerol, 80 mmol acetone Acetone/glycerol molar ratio = 2.	[231]
41	Nb_2_O_5_/HUSY zeolite	Acetone	3	40	solketal	66	98	65	Catalyst loading = 2 wt%; Acetone/glycerol molar ratio = 2.	[197]
36	Nb_2_O_5_ treated at (300 °C)	Acetone	6	70	solketal	80	92	74	Catalyst loading = 6.4 wt%; Acetone/glycerol molar ratio = 1.5.	[198]
42	10% PO_4_^3−^/8% Nb_2_O_5_/MCM	Lauric acid	5	110	glycerol monolaurate	96	93	89	Catalyst loading = 0.5 g; Glycerol/lauric acid molar ratio = 1.	[199]
1	20 Nb_2_O_5_ –SiO_2_	Water	2 10	320	acrolein	100 63	65 52	-	WHSV = 80 h^−1^ and feed = 30% solution of glycerol in water.	[238]
2	Nb-25/SBA-15 Nb_2_O_5_/SBA-15 Si/Nb-25/SBA-15	Water	4	350	acrolein	86 22 64	40 48 61	34 11 38	Catalyst loading = 0.1 g speed of glycerol dosing: 1 mL/h; Ar flow: 50 mL/min.	[213]
13	NbOPO_4_	Water	4–5	320	acrolein	100	87	87	0.20 g catalyst diluted with 3.0 g of quartz; 10 wt% glycerol in water; time on stream = 4–5 h; Flow rate = 0.5 g h^−1^.	[201]
15	50 wt%PO_4_/Nb_2_O_5_	Water	2	240	acrolein	68	72	49	Catalyst loading = 4 g; Feed composition: 10 wt% glycerol in water.	[200]
3	Nb/Zr oxides	Water	215	300	acrolein	100	71	71	Glycerol aqueous solution (20 wt%); flow rate: 3.8 g h^−1^; inert gas flow rate: 75 mL min^−1^; Time on stream after which the conversion decreased by 20%: 215 h.	[202]
4	Nb_2_O_5_ supported on Zr-doped silica	Water	8	235	acrolein	77	45	35	Catalyst weight = 0.50 g diluted with SiC to 3 cm^3^; Feed composition: 10 wt% glycerol in water; liquid flow: 0.1 mL min^−1^; N_2_ flow = 15 mL min^−1^; Time on stream = 8 h.	[203]
10	Nb,W-oxides supported on ZrO_2_-	Water	3	305	acrolein	100	75	75	Catalyst weight = 0.80 g; Feed composition: 20 wt% glycerol in water; liquid flow: 3.6 mL h^−1^; Ar flow = 15 mL min^−1^.	[204]
12	ZrNbO mixed oxides	Water	32	300	acrolein	100	70	70	Catalyst weight = 7.50 g; Feed composition: 20 wt.% glycerol in water; flow rate: 3.8 g h^−1^; inert gas flow rate = 75 mL min^−1^.	[205]
16	2.5 wt%PO_4_/W_2.8_Nb_2.2_O_14_	Water	1–2	285	acrolein	100	82	82	Catalyst weight/Flow rate = 2.5 × 10^−3^ g min mL^−1^; time on a stream: 1–2 h; water/glycerol molar ratio = 5; composition of reaction gas: Glycerol/O_2_/N_2_/H_2_O = 5/14/56/25 (mol%).	[188]
14	Siliconiobium phosphate (NbPSi-0.5)	Water	4	250	acrolein	100	76	76	Catalyst loading = 0.3 g; 10 wt% glycerol in water; composition of reaction gas: Glycerol/H_2_O/N_2_ = 1.3/53.6/40.2 (mol%); Flow rate = 2.1 mL h^−1^; GHSV = 14,940 mL g^−1^ h^−1^.	[214]
11	5% NbWO_x_/ZrO_2_	Water	3–4	290	acrolein	99	71	70	Catalyst loading = 0.3 g; GHSV = 1171 h^−1^; 60 mL min^−1^ N_2_ flow rate; time on a stream: 3–4 h; 10 wt% glycerol in water.	[206]
5	8 wt% Nb_2_O_5_ supported on Zr-doped silica and treated with H_3_PO_4_	Water	2	350	acrolein	100	74	74	Time on stream = 2 h; Catalyst loading: 0.5 g diluted with SiC to 3 cm^3^ volume; 10 wt% glycerol in water; liquid flow: 0.1 mL min^−1^; N_2_ flow = 15 mL min^−1^.	[207]
9	NbW-oxide on Al_2_O_3_	Water	3	305	acrolein	100	72	72	Catalyst loading = 0.8 g; Feed: 3.6 mL h^−1^; aqueous glycerol vaporized in 15 mL min^−1^ Ar; Time on stream = 3 h.	[208]
6	NbW-oxides	Water	1	285	acrolein	99	75	74	Catalyst weight = 0.20 g; Flow rate: 80 mL min^−1^; water/glycerol molar ratio = 5.	[209]
7	WNb oxides	Water		295	acrolein	100	82	82	glycerol/H_2_O/O_2_/He molar ratio of 2/40/4/54, contact time, W/F, of 81 gcat h (molgly)^−1^.	[117]
8	WNb oxides	Water	-	300	acrolein	100	83	83	81 g_cat_ h (mol_glycerol_)^−1^; glycerol/water/O_2_/He molar ratio = 2/40/4/54	[241]
18	Cs_2.5_H_0.5_PW_12_O_40_/Nb_2_O_5_	Water	9–10	320	acrolein	94	77	72	Glycerol concentration = 0.2 g g^−1^; N_2_/O_2_ flow rate = 18 mL min^−1^; N_2_/O_2_ = 5/1(L/L); time on stream = 9–10 h.	[211]
17	H_3_PW_12_O_40_/Nb_2_O_5_	Water	9–10	325	acrolein	100	89	89	Catalyst weight = 0.30 g; Feed = 0.5 mL h^−1^; N_2_ flow rate = 10 mL min^−1^; 10 wt.% glycerol in water.	[212]
19	Cs_2.5_H_0.5_PW_12_O_40_/Nb_2_O_5_	Water	10	300	acrolein	96	83	80	Catalyst loading = 0.50 g; glycerol/H_2_O/N_2_/O_2_ molar ratio = 1/2/68/12; flow rate = 0.6 mL h^−1^; 20 wt% glycerol in water; time on stream = 10 h.	[215]
44	Pt/Nb_2_O_5_	Water	1	140	1,2-propanediol	50	88	44	P_H2_ = 50 bar; Catalyst loading = 0.75 g; 1.50 g Amberlyst 15; 20% glycerol in water.	[220]
45	Pt/Nb-WO_x_	Water	12	160	1,3-propanediol	50	28	14	P_H2_ = 5 MPa; Catalyst loading = 0.30 g; 5 wt% glycerol in water (12 mL).	[215]
46	Ni/Nb_2_O_5_/Al_2_O_3_	Water	30	500	H_2_	90	60	54	Catalyst loading = 0.15 g; Water/glycerol ratio = 16	[218]

^a^ MPTMS (MP) = (3-mercaptopropyl)trimethoxysilane. MCF = mesostructured cellular foams, SBA-15 = hexagonally ordered mesoporous silica. ^b^ time on stream during which activity and selectivity was stable.

## 4. Conclusions and Outlook

It has been shown that Nb-based materials show excellent performance as catalysts for the conversion of biomass, which is a very important application to guarantee a better future for human beings. Niobium phosphate is one of the most promising materials to convert cellulose into different products, especially when coupled with noble metals such as Ru or Pt. On the other hand, Nb_2_O_5_ has been the most used niobium catalyst/support in lignin conversion. NbOPO_4_ and Nb_2_O_5_ are also efficient for converting glucose or fructose into HMF. The conversion of xylose to furfural has been conducted more widely using Nb_2_O_5_ catalysts. Niobium phosphate and niobium oxide have also been the most often used niobium catalysts for converting glycerol, a by-product of biodiesel production, into chemicals with added commercial value. Although significant advances have been made in recent years in developing niobium-based catalysts for biomass conversion reactions, the control of acid–base properties, morphology, and specific surface area are still challenging topics that need to be studied. Another important point concerns the relatively high temperatures used in the biomass conversion reactions, ranging from 150 to 350 °C. Alternative strategies to conventional catalytic processes to lower those temperatures include photocatalytic, photoelectrochemical, and electrochemical processes that are still timidly used but may play a fundamental role shortly. In this sense, future studies may also focus on improving the optical properties of niobium compounds to increase light absorption in the visible spectrum region when the objective is to use solar energy.

However, the niobium is versatile and can be used for other purposes of similar importance to the above-mentioned ones. An example is a fabric with visible light photocatalytic activity made of polycaprolactone and iron niobate nanoparticles for the destruction of methyl paraoxon in visible radiation [109]. Methyl paraoxon is an organophosphate with acute toxicity: a perfect molecule to simulate a chemical warfare nervous agent. The material almost completely degraded the methyl paraoxon after 48 h under visible light, generating compounds that are less harmful to human health, opening the possibility of use in soldiers’ clothing for chemical warfare protection. Applied in its polyoxometalate form, niobium has phototherapeutic effects against tumor cells. In the Maia et al. study [42], polyoxoniobate carrying peroxo species (O-O) showed inhibitory activity on leukemia (in vitro) when exposed to less harmful UV radiation (UV-A). These oxidizing species present in the complex improve the affinity with DNA and when exposed to this type of light, they favor the inviability of the cell death. Thus, niobium compounds are fascinating materials which can be applied not only as efficient catalysts in the production of valuable chemicals but also can be used in many fields important for human life.

## Figures and Tables

**Figure 1 molecules-28-01527-f001:**
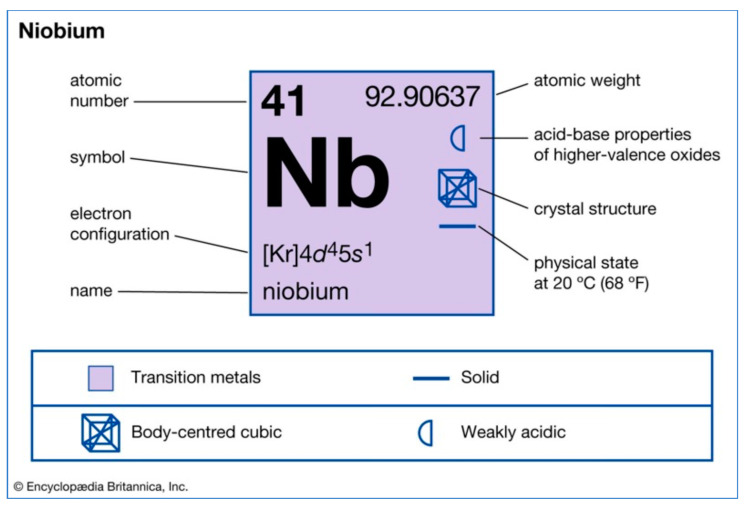
Characteristics of the element niobium.

**Figure 2 molecules-28-01527-f002:**
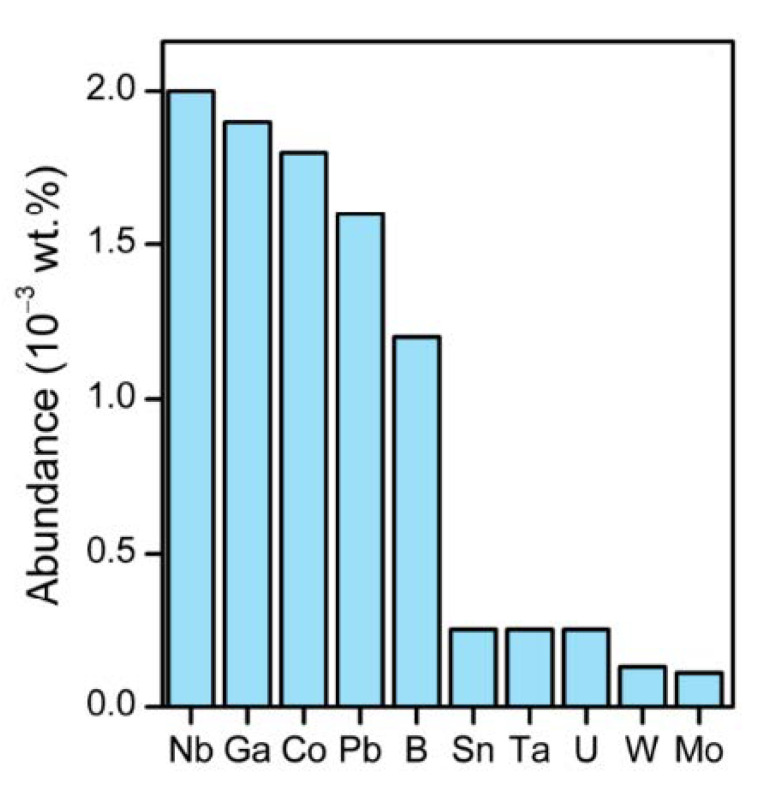
Elements less abundant than niobium in the earth’s crust. The data were obtained with permission from reference [10]. Copyright 2014 Royal Society of Chemistry.Currently, three companies explore niobium in Brazil: the Brazilian Company of Metallurgy and Mining (CBMM), Anglo American Brazil Ltd.@ (Mineração Catalão Goiás), and Mineração Taboca (Paranapanema Group). Brazil contains 98% of the known reserves of niobium in the world, which corresponds to 842.46 million tons [8,9]. The largest Brazilian reserves of niobium are accumulated in pyrochlore ores in the Araxá city, state of Minas Gerais. The ferroniobium alloy is produced through a partnership, of which Minas Gerais Economic Development Company (Codemig) is a participating partner and CBMM is an ostensible partner, whose majority shareholder is the Moreira Salles group.

**Figure 3 molecules-28-01527-f003:**
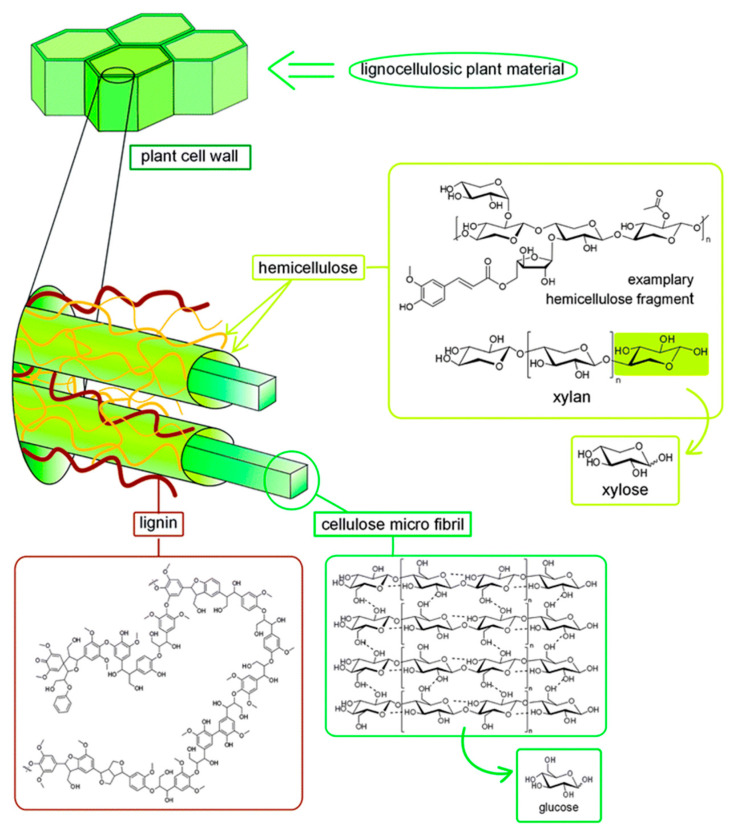
Composition of lignocellulosic biomass. Reproduced with permission from ref. [80]. Copyright 2013 Royal Society of Chemistry.

**Figure 4 molecules-28-01527-f004:**
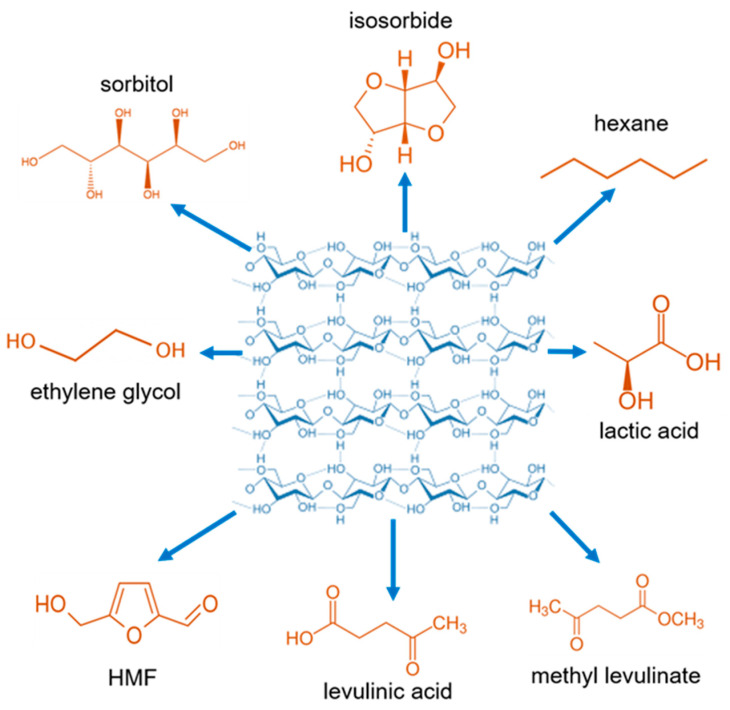
Conversion of cellulose into different molecules catalyzed by niobium compounds.

**Figure 5 molecules-28-01527-f005:**
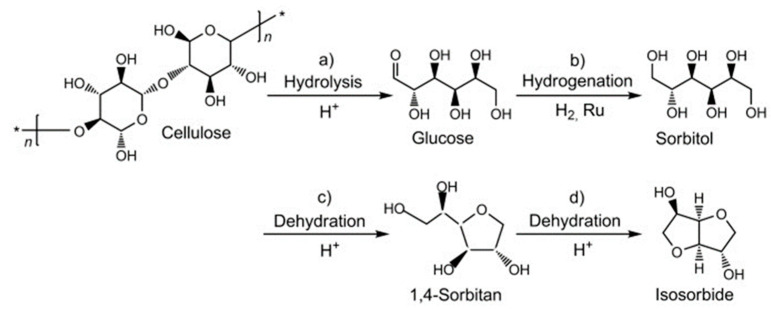
Steps for converting cellulose into isosorbide. Reproduced with permission from ref. [99] Copyright © 2013 WILEY-VCH Verlag GmbH & Co.

**Figure 6 molecules-28-01527-f006:**
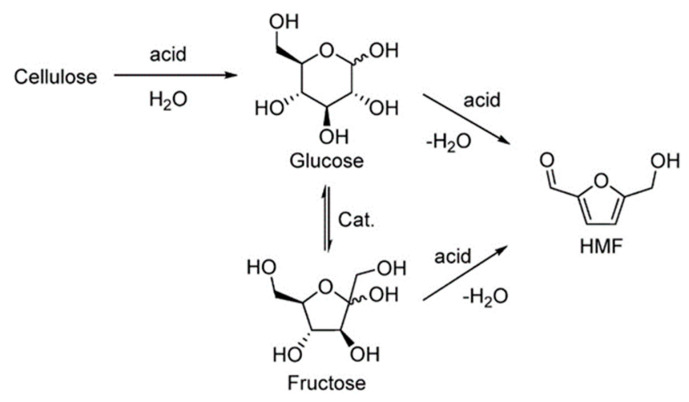
Cellulose conversion into 5-hydroxymethylfurfural. Adapted with permission from ref. [102] © 2018 Elsevier B.V. All rights reserved.

**Figure 7 molecules-28-01527-f007:**
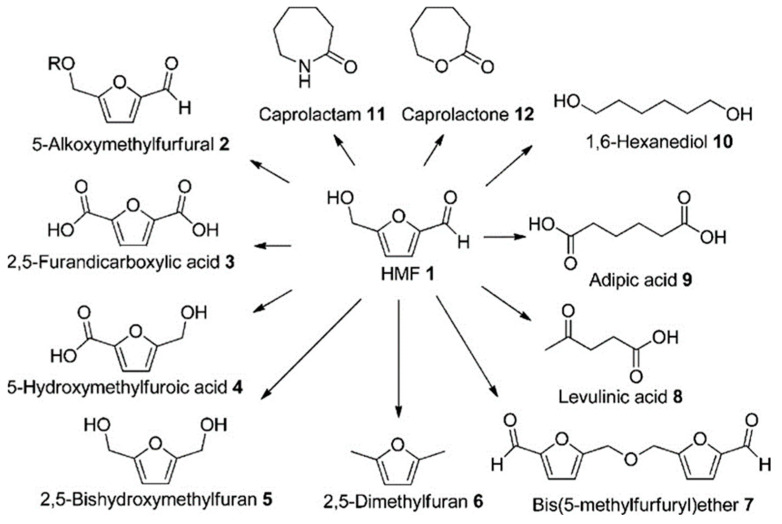
5-Hydroxymethylfurfural as a platform molecule. Reproduced with permission from ref. [103] Copyright © 2013 American Chemical Society.

**Figure 8 molecules-28-01527-f008:**
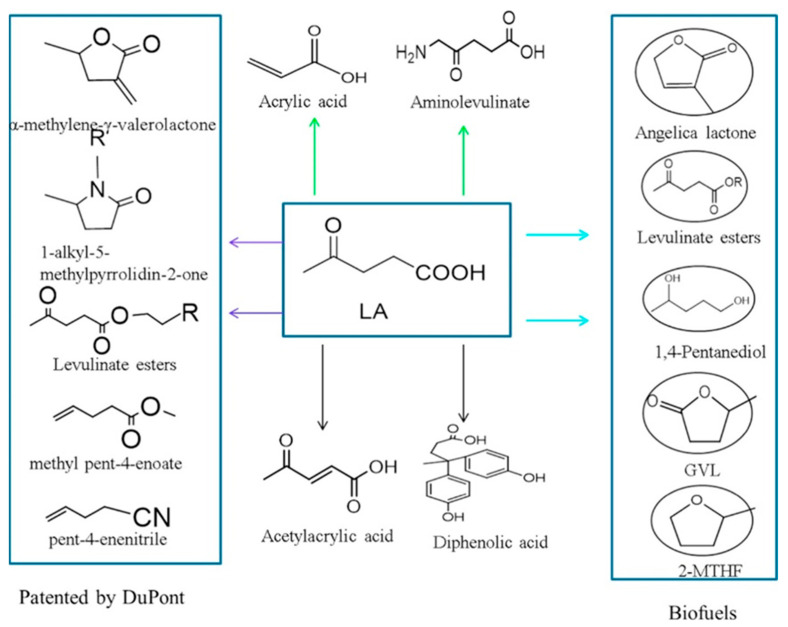
Levulinic acid as platform molecule for value-added chemicals. Reproduced with permission from ref. [108] Copyright © 2015 Elsevier Ltd. All rights reserved.

**Figure 9 molecules-28-01527-f009:**
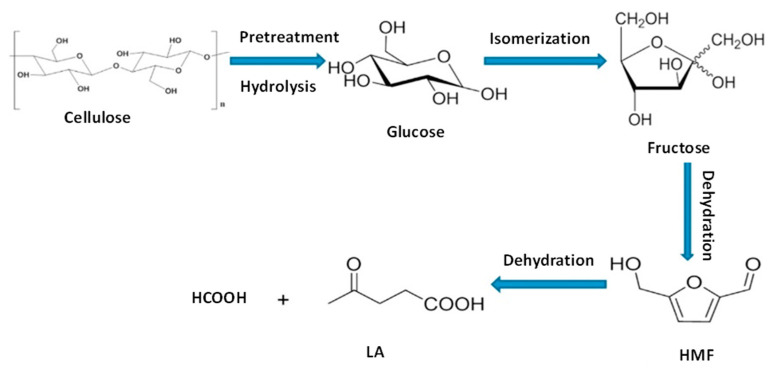
Conversion of cellulose into levulinic acid (LA). Reproduced with permission from ref. [108] Copyright © 2015 Elsevier Ltd. All rights reserved.

**Figure 10 molecules-28-01527-f010:**
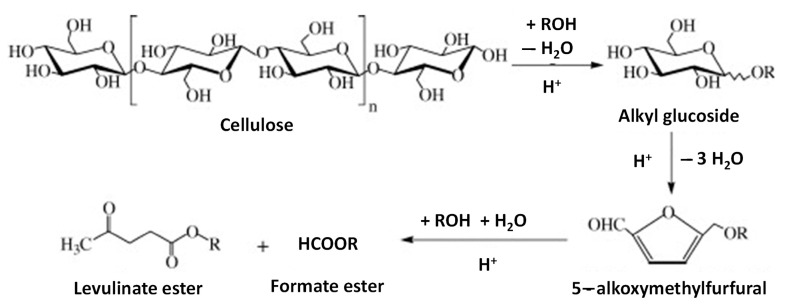
Conversion of cellulose into levulinate esters. Reproduced with permission from ref. [113] Copyright © 2011 Elsevier Ltd.

**Figure 11 molecules-28-01527-f011:**
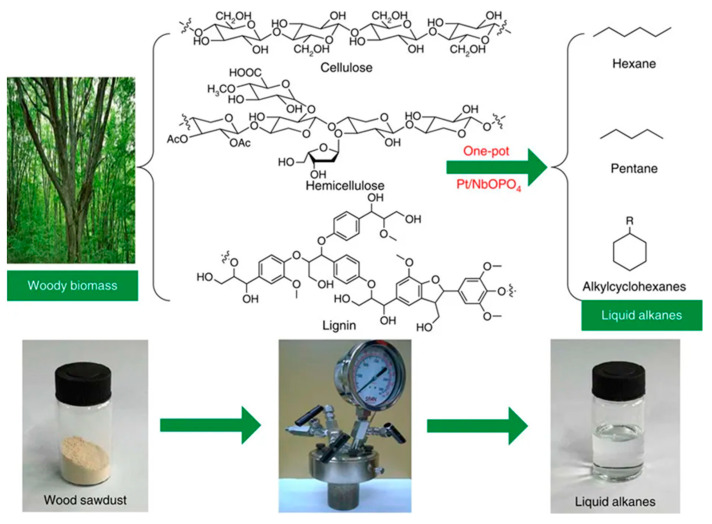
Direct conversion of wood cellulose, hemicellulose, and lignin fractions into hexane, pentane, and alkylcyclohexane, respectively, in the presence of NbOPO_4_/Pt catalyst in cyclohexane. Reproduced with permission from ref. [113] Copyright © 2016 Nature.

**Figure 12 molecules-28-01527-f012:**
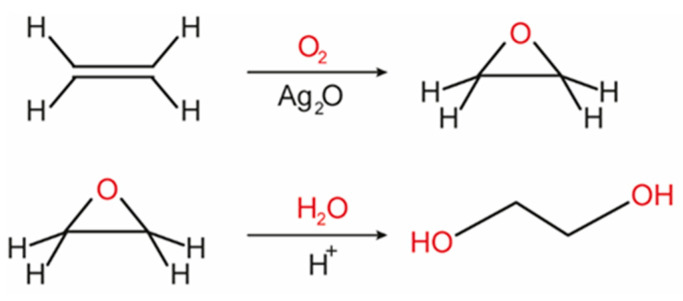
Industrial synthesis of ethylene glycol from ethylene gas.

**Figure 13 molecules-28-01527-f013:**
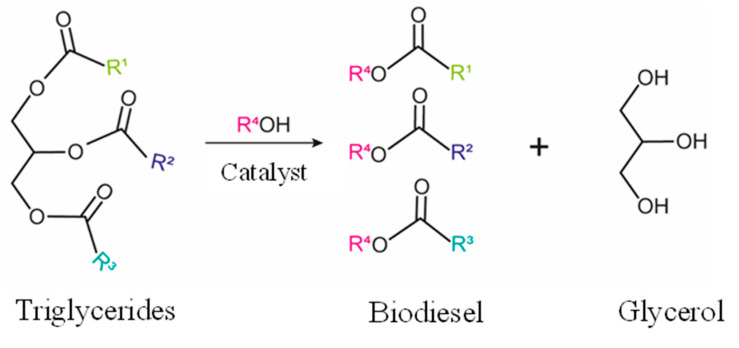
Transesterification of triglycerides for biodiesel production (R1, R2, R3 and R4 are alkyl groups).

**Figure 14 molecules-28-01527-f014:**
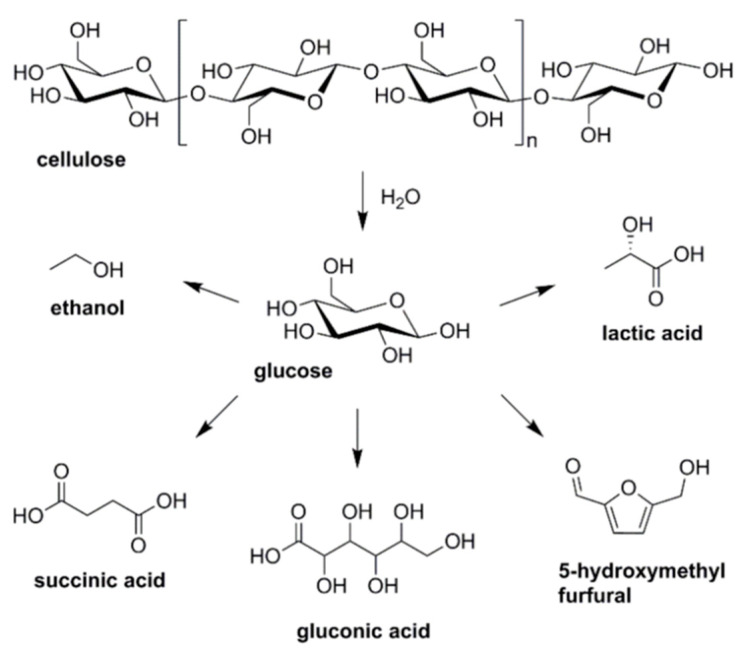
Proposed scheme for cellulose hydrolysis reaction showing some of the products that can be obtained by glucose transformation. Reproduced with permission from ref. [135] Copyright © 2019 John Wiley and Sons.

**Figure 15 molecules-28-01527-f015:**
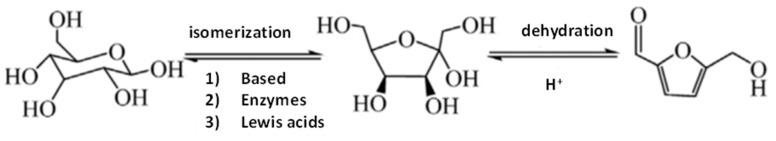
Synthesis of HMF from glucose. Reproduced with permission from ref. [90] Copyright © 2017 Elsevier.

**Figure 16 molecules-28-01527-f016:**
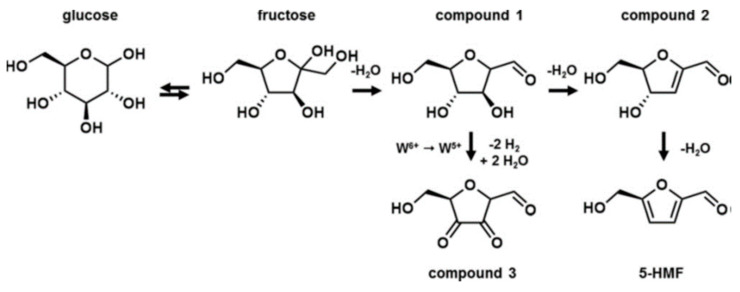
Reaction intermediates in the conversion of glucose to HMF. Reproduced with permission from ref. [140] Copyright © 2016 Wiley Online Library.

**Figure 17 molecules-28-01527-f017:**
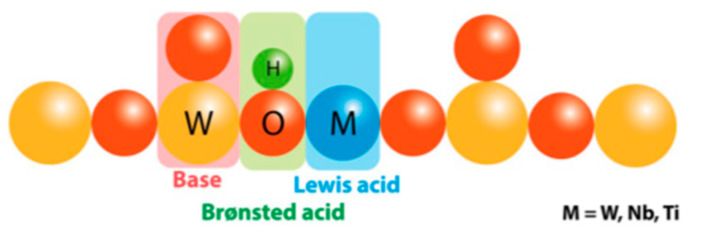
Active centers in MWO3 catalytic systems. Reproduced with permission from ref. [141] Copyright © 2018 Springer.

**Figure 18 molecules-28-01527-f018:**
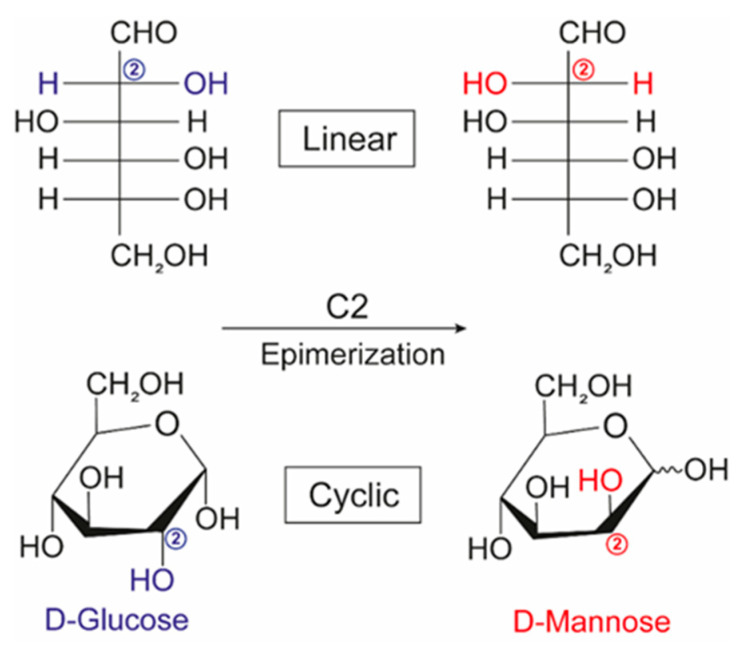
Scheme showing the difference between D-glucose and D-mannose in linear and cyclic form.

**Figure 19 molecules-28-01527-f019:**
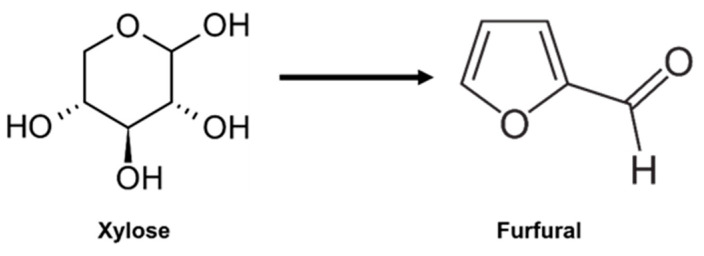
Conversion of xylose into furfural.

**Figure 20 molecules-28-01527-f020:**
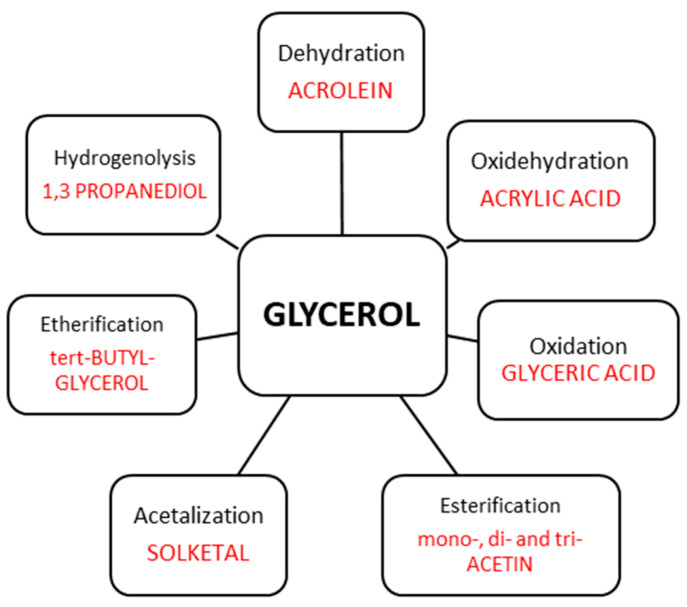
Transformation of glycerol in catalytic reactions performed on niobium-containing catalysts.

**Figure 21 molecules-28-01527-f021:**
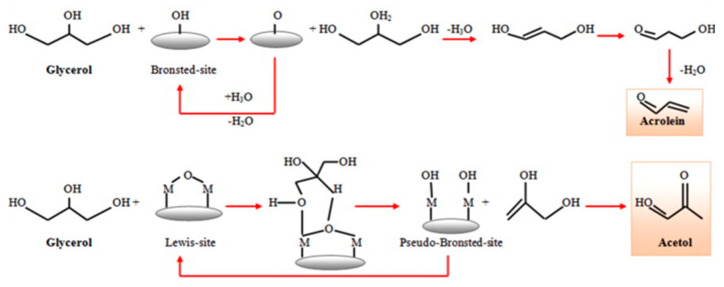
Catalytic glycerol dehydration over Brønsted and Lewis acid sites (adapted from [191]).

**Figure 22 molecules-28-01527-f022:**
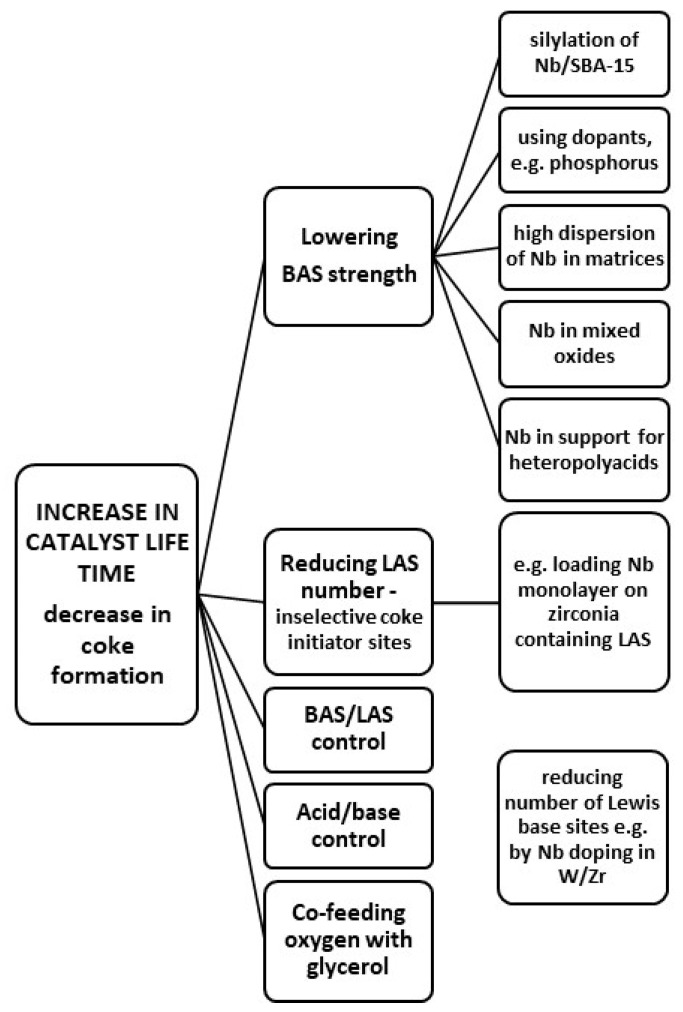
Selected ways leading to the increase in stability of catalysts containing niobium in glycerol dehydration.

**Figure 23 molecules-28-01527-f023:**
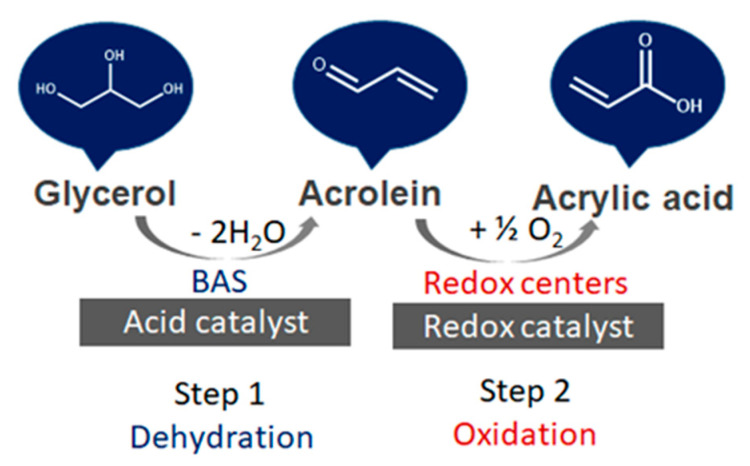
Pathway of the glycerol oxidative dehydration to acrylic acid.

**Figure 24 molecules-28-01527-f024:**
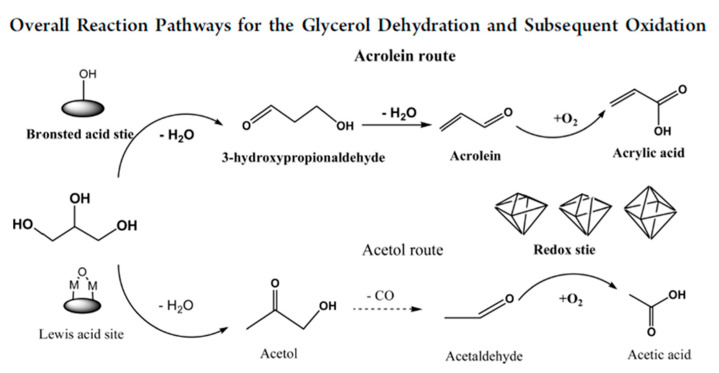
Two different reaction routes in the oxidative dehydration of glycerol depending on the nature of acid sites on the catalyst surface. Reproduced with permission from ref. [220] Copyright © 2018 Elsevier.

**Figure 25 molecules-28-01527-f025:**
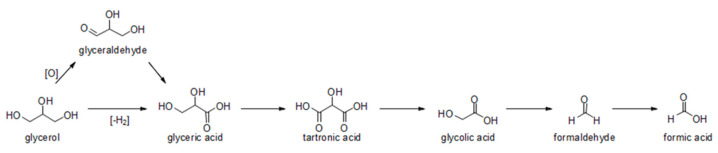
The scheme of glycerol oxidation with oxygen in liquid phase. Reproduced with permission from ref. [224] Copyright © 2009 Elsevier.

**Figure 26 molecules-28-01527-f026:**
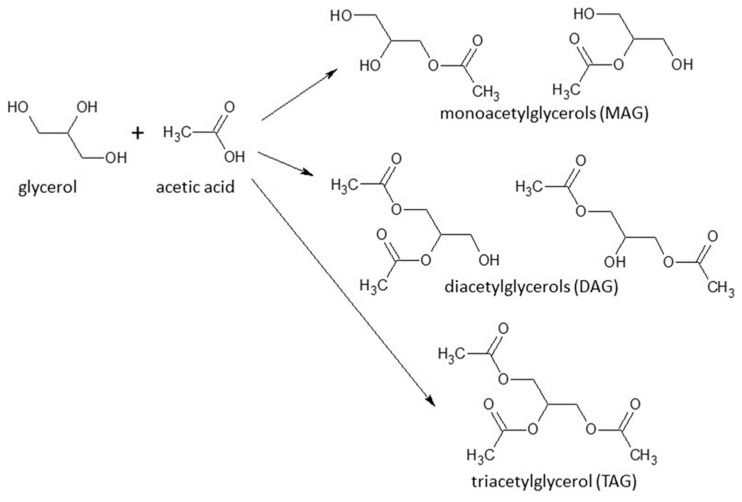
Esterification of acidic acid with glycerol. Reproduced with permission from ref. [192] Copyright © 2011 Elsevier.

**Figure 27 molecules-28-01527-f027:**
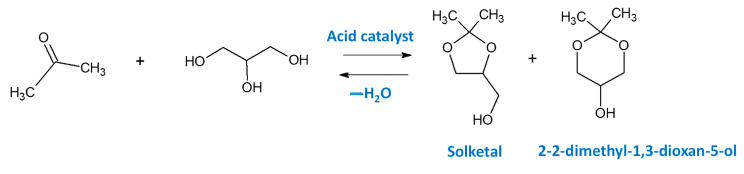
Acetalization of glycerol with acetone over acid catalysts. Adapted with permission from ref. [231]. Copyright © 2014, American Chemical Society.

**Figure 28 molecules-28-01527-f028:**
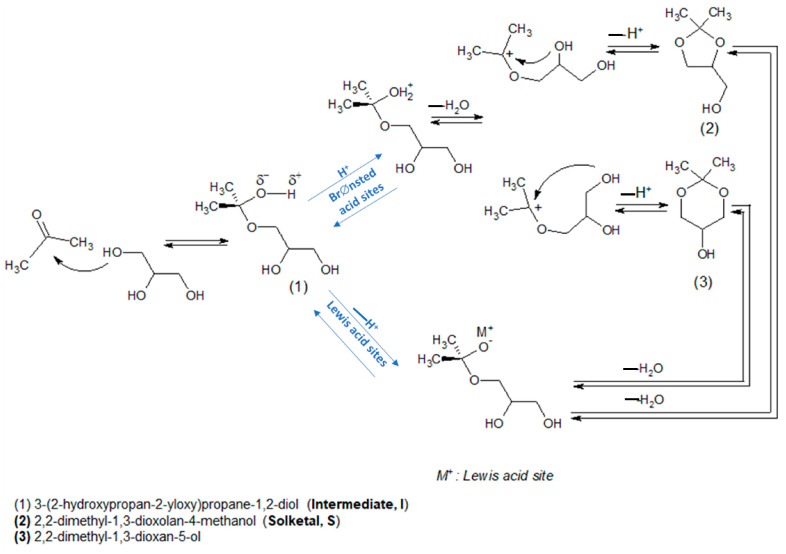
Reaction mechanism proposed for the acetalization of glycerol with acetone over acid catalyst. Adapted with permission from [231] Copyright © 2014, American Chemical Society.

**Table 1 molecules-28-01527-t001:** Review papers on the synthesis, characterization, and catalytic applications of niobium compounds over the last 30 years.

Entry	Year	Authors	Title	Notes	Ref.
1	1995	Tanabe and Okazaki	Various reactions catalyzed by niobium compounds and materials	Review on the use of niobium compounds (e.g., Nb_2_O_5_, Nb_2_O_5_·H_2_O, NbOPO_4_, K_4_Nb_6_O_17_, HCa_2_Nb_3_O_10_, and composites) in catalytic reactions (e.g., dehydration of alcohols, hydration of olefins, esterification, hydrolysis, condensation, alkylation, etc.). 116 references	[68]
2	1999	Nowak and Ziolek	Niobium compounds: preparation, characterization, and application in heterogeneous catalysis	Review on niobium chemistry including preparation, structure, and physicochemical and catalytic properties of niobium compounds for dehydration of alcohols, dehydrogenation, oxidative dehydrogenation, oxidation and ammoxidation, esterification, alkylation, isomerization, hydrogenolysis, hydrogenation. NO reduction, etc. 187 references	[69]
3	2003	Ziolek	Niobium-containing catalysts—the state of the art	Review on the use of niobium compounds (e.g., oxides_,_ sulfides, nitrides, oxynitrides, carbides, oxycarbides, and phosphates) in heterogeneous catalysis (e.g., liquid and gas phase oxidation). 131 references.	[70]
4	2003	Tanabe	Catalytic application of niobium compounds	Review on catalytic applications of niobium compounds in oxidative dehydrogenation of alkanes, oxidative coupling of methane, oxidation and ammoxidation, removal of nitrogen oxides, etc. 63 references.	[71]
5	2006	Andrade and Rocha	Recent applications of niobium catalysts in organic synthesis	Mini-review on applications of niobium compounds in catalytic organic synthesis (e.g., Biginelli reactions, Friedel–Crafts acylation and Sakurai–Hosomi reactions of acetals, Knoevenagel condensation, acetylation of alcohols and phenols, etc. 28 references.	[72]
6	2009	Guerrero-Perez and Bañares	Niobium as promoting agent for selective oxidation reactions	Brief revision on the promoting effect of niobium on different catalytic reactions (e.g., oxidative dehydrogenation lower alkanes, oxidation of ethane to acetic acid, oxidation and ammoxidation of propane, methane partial oxidation, oxidation of n-butane to maleic anhydride, and oxidative coupling of methane). 76 references	[73]
7	2012	Nowak	Frontiers in mesoporous molecular sieves containing niobium: From model materials to catalysts	Review on mesoporous molecular sieves containing niobium and its application in liquid and gas phase oxidation. 81 references.	[74]
8	2012	Zhao et al.	Nanostructured Nb_2_O_5_ catalysts	Review on synthetic methods for the preparation of Nb_2_O_5_ nanostructures and their potential applications in catalysis. 58 references.	[75]
9	2016	Nico et al.	Niobium oxides and niobates physical properties: review and prospects	Review on stoichiometric and non-stoichiometric phases of niobium oxides (e.g., NbO, NbO_2_, Nb_2_O_5_, and Nb_2_O_5-δ_) and niobates (alkali niobates, columbite niobates, and rare earth niobates). 240 references.	[76]
10	2017	Ziolek and Sobczak	The role of niobium component in heterogeneous catalysts	Review on the role of niobium supported zeolites in the enhancement of the redox properties of Cu, Ag, Au, and Pt, acidic and basic properties, and catalytic properties for the oxidation of cyclohexene, glycerol, and methanol. 124 references.	[12]
11	2018	Yan et al.	Liquid-phase epoxidation of light olefins over W and Nb nanocatalysts	Perspective on liquid-phase epoxidation of light olefins (ethylene, propylene, hexene, octene) on Nb and W catalysts. 256 references	[77]
12	2018	Wawrzynczak et al.	Toward exploiting the behavior of niobium-containing mesoporous silicates vs. polyoxometalates in catalysis	Review describes the catalytic behavior of niobium–polyoxometalates in water oxidation reaction, H_2_ evolution reaction, epoxidation reaction, base catalysis, and the catalysis on heterogeneous niobium-containing mesoporous silicates (e.g., oxidation of unsaturated compounds and Knoevenagel condensation). 154 references.	[78]
13	2019	Siddiki et al.	Lewis acid catalysis of Nb_2_O_5_ for reactions of carboxylic acid derivatives in the presence of basic inhibitors	Review on the use of Nb_2_O_5_ and Nb_2_O_5_·H_2_O in nucleophilic substitution reactions of carboxylic acid derivatives. 95 references	[79]

**Table 2 molecules-28-01527-t002:** Performance of niobium-based catalysts for conversion of cellulose into different products—from 2010 to date.

Entry	Catalyst	Solvent	t (h)	T (°C)	Product	C (%)	S (%)	Y (%)	Experimental Conditions	Ref.
1	NbOPO_4_/Ru (pH 2)	water	24	160	sorbitol	100	69	69	0.24 g cellulose mixed with 0.1 g catalyst and ball-milled for 10 h, 30 mL water, 4 MPa H_2_, Ru loading = 5 wt%; pH = 2	[85]
2	NbOPO_4_/Ru (pH 2)	water	24/18	170/230	isosorbide	-	-	57	First step: 0.24 g cellulose, 15.0 g H_2_O and 0.1 g Ru/NbOPO_4_, 170 °C, 4 MPa H_2_ for 24 h; Second step: the cellulose-derived sorbitol and sorbitan aqueous; 0.1 g NbOPO_4_, 230 °C, 1 MPa N_2_ for 18 h. Ru loading = 5 wt%	[86]
3	NbOPO_4_ (pH 7)	MIBK/ water	1	140	HMF	-	-	16	0.5 g cellulose, 10 mL of reaction solution, MIBK/ Water = 70:30, *v/v*. MIBK = methyl isobutyl ketone	[87]
4	NbO_x_/ZrO_2_	water	5	200	HMF	52	31	16	10 g L^−1^ cellulose, 10 g L^−1^ catalyst, volume = 45 mL, P_Ar_ = 10 bar, 1500 rpm (pH 3.6)	[88]
5	Nb_2_O_5_	water	3	200	HMF	-	-	19	10 wt% celulose; 1 wt% catalyst	[89]
6	Nb-SBA15	THF/H_2_O	8	170	HMF	94	54	51	0.10 g catalyst; 0.10 g cellulose; 6 mL THF + 2 mL H_2_O saturated with NaCl	[90]
7	Nb_2_O_5_/C	THF/H_2_O	8	170	HMF	91	58	53	0.1 g cellulose, 0.2 g Nb/C-50 catalyst, 2 mL water saturated with NaCl, 6 mL THF	[91]
8	25Nb@AlF_3_	water	2	180	Lactic acid	43	40	17	5 mL water, 0.16 g cellulose, 0.06 g catalyst	[92]
9	Preboiled 25Nb@AlF_3_	water	2	180	Lactic acid	34	79	27	5 mL of water, 0.16 g of cellulose, 0.06 g of catalyst, 1000 rpm	[93]
10	Al-NbOPO_4_	water	24	180	Levulinic acid	>90	-	53	0.5 g cellulose, 0.4 g catalyst, 10 g H_2_O	[94]
11	NbOPO_4_ (pH 1)	Methanol	24	180	Methyl levulinate	98	58	57	0.5 g of cellulose, 0.0004 mol of acid catalysts, 10 g of 95% methanol	[95]
12	Mesoporous Nb-doped WO_3_	water	3	225	Ethylene glycol	95	40	38	0.3 g cellulose, 0.05 g retro-aldol catalyst, 0.01 g 5 wt% Ru/C (1 wt% Ru w.r.t. WO_3_), 40 mL H_2_O, 45 bar H_2_, 600 rpm	[96]
13	HNbMoO_6_	-	4	25	Water-soluble sugars	100	72	72	microcrystalline cellulose (Avicel 0.4 g), solid additive (0.4 g), 600 rpm	[97]
14	NbOPO_4_/Pt	cyclohexane	20	190	hexane	-	-	72	5 MPa H_2_; 0.2 g cellulose, 0.2 g catalyst, and 6.46 g cyclohexane	[98]

**Table 3 molecules-28-01527-t003:** Performance of niobium-based catalysts for conversion of lignin into different products—from 2010 up to date.

Entry	Catalyst	Solvent	t (h)	T (°C)	Product	C (%)	S (%)	Y (%)	Experimental Conditions	Ref.
1	4 wt% Co@Nb_2_O_5_@Fe_3_O_4_	Water	6	180	C_20_–C_28_ and C_29_–C_37_ fragments	53	96	51	10 atm H_2_; 0.01 g catalyst, 0.4 g lignin, 5 mL solvent	[95]
2	Fe_3_O_4_@Nb_2_O_5_@Co@Re	Water	6	180	C_29_–C_37_ fragments	~85	~100	85	0.02 g catalyst, 0.01 g lignin, 2.5 mL H_2_O, 10 bar H_2_	[116]
3	Nb_2_O_5_/Ru	Water/cyclohexane	20	250	C_7_–C_9_ arenes	99	88	87	0.1 g lignin, 0.2 g catalyst, 10 mL H_2_O, 5 mL cyclohexane, 0.5 MPa H_2_	[118]
4	Nb_2_O_5_/Ru	Water	20	250	C_7_–C_9_ arenes	~50	95	~50	0.1 g enzymatic lignin, 0.2 g catalyst, 15 mL H_2_O, initial H_2_ pressure of 0.7 MPa	[126]
5	Nb_2_O_5_/Ru	Water/cyclohexane	20	250	C_7_–C_9_ arenes	38	86	33 (wt.%)	lignin oil (0.1 g, birch wood), 2% Ru/Nb_2_O_5_ (0.2 g), iPrOH (1.0 g), H_2_O (14 mL), cyclohexane (5 mL), 20 h	[119]
6	Nb_2_O_5_/Ru	Water	20	250	C_7_–C_9_ arenes	32 wt.%	71 wt.%	23 wt.%	0.1 g of birch lignin, 0.2 g catalyst, 15 mL water, 0.7 MPa H_2_, 2 wt% Ru	[121]
7	Zr-doped NbOPO_4_/Ni	C_10_	5.5	220	oxygen-free aromatics	84	62	52	8 wt% diaryl ether in 20 mL C10, 0.1 g catalyst, 0.5 MPa, 700 rpm	[120]
8	Nb_2_O_5_	-	2	60	-	45	-	-	Catalyst loading: 1 g catalyst/1 g switchgrass	[127]
9	NbN	Supercritical ethanol	1	340	Aromatic monomer	-	-	17 (wt.%)	200 mg catalyst, 400 mg lignin, 20 mL solvent, 10 bar H_2_.	[122]
10	CH_2_Cl_2_-modified Nb_2_O_5_/Ru	Water	20	250	Indane and its derivatives	34	70	24 (wt.%)	0.10 g catalyst, 0.10 g lignin-oil, 15 mL water, 0.5 MPa H_2_, 700 rpm	[123]
11	Nb_2_O_5_	Water	2	90	vanillin	-	-	-	0.50 g catalyst, 100 mL water, 8 g lignin, 150 rpm. Produced vanillin = 137 mg L^−1^	[124]
12	Ni_0.92_Nb_0.08_O	C_10_	2	200	cyclohexane	>99	100	-	0.1g Ni_0.92_Nb_0.08_ catalyst, 8 wt% substrate, 20 mL C10, 3 MPa, 700 rpm	[125]
13	Pd/Nb_2_O_5_/SiO_2_	Cyclohexane	24	170	cyclohexane	100	98	98	catalyst (0.2 g), diphenil ether (0.2 g), cyclohexane (6.46 g), 2.5 MPa H_2_	[128]
14	Nb_2_O_5_	Water	5	60	monomeric phenolic compounds			47 wt.%	0.2–1 g lignin/g peracetic acid; 10 wt% Nb_2_O_5_	[129]
15	Pt/NbOPO_4_	Cyclohexane	20	190	alkylcyclohexanes	-	-	36	0.4 g of birch sawdust, 0.4 g of catalyst, and 6.46 g of cyclohexane, and 5 MPa H_2_	[98]

## Data Availability

No new data were created or analyzed in this study. Data sharing is not applicable to this article.

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
