# Peer review of "Niobium: The Focus on Catalytic Application in the Conversion of Biomass and Biomass Derivatives"

_molecules, 2023, doi:10.3390/molecules28041527_

Round 1
Reviewer 1 Report
1. It is suggested that go straight into the topic and start with the introduction of Nb. If possible, please merge 1.1 and 1.2 and delete the movie-related content in 1.1, and reduce the numbers of paragraphs.
2. Please replace Figure 1 by the pictures of Nb atomic structures.
3. Where is the caption of Table 1?
4. Please improve the resolutions of Figure 22.
5. It is suggest that move Table 2&3&4&5&6&7 to the supporting imformations.
6. Delete the movie-related content in the conclusion and summarize Nb again.
Author Response
Referee #1
Open Review
(x) I would not like to sign my review report
( ) I would like to sign my review report
English language and style
( ) English very difficult to understand/incomprehensible
( ) Extensive editing of English language and style required
(x) Moderate English changes required
( ) English language and style are fine/minor spell check required
( ) I don't feel qualified to judge about the English language and style
|
Is the work a significant contribution to the field? |
|
|
Is the work well organized and comprehensively described? |
|
|
Is the work scientifically sound and not misleading? |
|
|
Are there appropriate and adequate references to related and previous work? |
|
|
Is the English used correct and readable? |
Comments and Suggestions for Authors
- It is suggested that go straight into the topic and start with the introduction of Nb. If possible, please merge 1.1 and 1.2 and delete the movie-related content in 1.1, and reduce the numbers of paragraphs.
As the Black Panther saga is inspired by a niobium mine in the Congo, as well as the versatility of using vibranium and niobium, we understood that there was a playful context for the comparison. However, we accept the reviewer's suggestion.
- Please replace Figure 1 by the pictures of Nb atomic structures.
The reviewer's suggestion was followed.
- Where is the caption of Table 1?
The Table 1 caption has been added.
- Please improve the resolutions of Figure 22.
I have included the improved Fig. 22 to the attached manuscript an to the end of this file.
- It is suggest that move Table 2&3&4&5&6&7 to the supporting imformations.
In this case, we decided not to follow the reviewer's suggestion because we consider the information in the Tables to be important for understanding the manuscript.
- Delete the movie-related content in the conclusion and summarize Nb again.
The reviewer's suggestion was followed.
Reviewer 2 Report
Congratulations for the article
There is an exhaustive review study, with a considerable number of updated references. Very clear explanation of the concepts. It is quite a complete article.
Well structured, very clear. The review of the conversion of polymers is very complete.
The conclusions are giving an account of what was exposed during the text and a good presentation of the perspectives is made.
Author Response
Referee #2
Open Review
(x) I would not like to sign my review report
( ) I would like to sign my review report
English language and style
( ) English very difficult to understand/incomprehensible
( ) Extensive editing of English language and style required
( ) Moderate English changes required
( ) English language and style are fine/minor spell check required
(x) I don't feel qualified to judge about the English language and style
|
Is the work a significant contribution to the field? |
|
|
Is the work well organized and comprehensively described? |
|
|
Is the work scientifically sound and not misleading? |
|
|
Are there appropriate and adequate references to related and previous work? |
|
|
Is the English used correct and readable? |
Comments and Suggestions for Authors
Congratulations for the article
There is an exhaustive review study, with a considerable number of updated references. Very clear explanation of the concepts. It is quite a complete article.
Well structured, very clear. The review of the conversion of polymers is very complete.
The conclusions are giving an account of what was exposed during the text and a good presentation of the perspectives is made.
The authors are grateful for the reviewer's comments.
Reviewer 3 Report
The review focuses on the catalytic conversion of biomass or biomass derivatives and is organized into two main chapters: Conversion of Plant Biomass Fractions over Niobium-Containing Catalysts (Chap 2) Conversion of Plant Biomass Derivatives over Niobium-Containing Catalysts (Chap 3). The review is well-focused on a well-defined field, i.e. the use of Niobium as a metal with catalytic properties for biomass conversion, and analyses a fair number of compounds, catalysts and reactions, including rather detailed and complete tables, so I think that its publication may be useful to researchers working in this field. The following minor revisions are suggested:
1) 1.1. Motivation: Vibranium from Marvel’s stories.
I am sorry, but I did not appreciate the motivation given in this paragraph. The reference to a science fiction saga did not involve me or arouse particular interest. I think it does not improve the quality of the paper; indeed I consider it inappropriate; and particularly Figure 1 seems to me not appropriate for a scientific paper. My suggestion is to remove figure 1, this "original" motivation for research, and the relevant comments in the conclusions. In any case, this comment expresses my personal opinion and I leave the final decision about this to the editor.
2) Figure 2: which is the good reason to show graph (b)? It reports data apparently not relevant to the text. Give some valid reasons to show it or remove it!
3) Lines 207-215 “As no comprehensive review of the use of niobium catalysts in biomass conversion has been published yet, we would like to fill this gap with presenting a review of the use of niobium catalysts for converting biomass fractions such as cellulose, hemicellulose, and lignin into initial platform molecules for high value-added products. The catalytic conversion of plant biomass lipids by niobium compounds as well as the conversion of glycerol, the byproduct derived from those reactions, into value-added molecules will also be discussed. The proposed review fills the gap in the review papers devoted to niobium catalysts because it focuses on the catalytic reactions (conversion of biomass and biomass derivatives) not covered yet in any review article”
I suggest you review this paragraph, which brings up the same concepts several times. I strongly suggest making it leaner, more concise and clearer, and less redundant.
4) The review is organized into two main chapters: Chap 2 and Chap 3. Each chapter is subdivided into paragraphs and sub-paragraphs each dedicated to a reaction of one or two biomass species or derivatives. Being the manuscript is quite long, it appears a bit disorienting for the reader: I suggest adding a short index between paragraph "1.4-Scope of the review" and chapter 2 to improve the consulting facility.
Author Response
Referee #3
Open Review
(x) I would not like to sign my review report
( ) I would like to sign my review report
English language and style
( ) English very difficult to understand/incomprehensible
( ) Extensive editing of English language and style required
( ) Moderate English changes required
(x) English language and style are fine/minor spell check required
( ) I don't feel qualified to judge about the English language and style
|
Is the work a significant contribution to the field? |
|
|
Is the work well organized and comprehensively described? |
|
|
Is the work scientifically sound and not misleading? |
|
|
Are there appropriate and adequate references to related and previous work? |
|
|
Is the English used correct and readable? |
Comments and Suggestions for Authors
The review focuses on the catalytic conversion of biomass or biomass derivatives and is organized into two main chapters: Conversion of Plant Biomass Fractions over Niobium-Containing Catalysts (Chap 2) Conversion of Plant Biomass Derivatives over Niobium-Containing Catalysts (Chap 3). The review is well-focused on a well-defined field, i.e. the use of Niobium as a metal with catalytic properties for biomass conversion, and analyses a fair number of compounds, catalysts and reactions, including rather detailed and complete tables, so I think that its publication may be useful to researchers working in this field. The following minor revisions are suggested:
1) 1.1. Motivation: Vibranium from Marvel’s stories.
I am sorry, but I did not appreciate the motivation given in this paragraph. The reference to a science fiction saga did not involve me or arouse particular interest. I think it does not improve the quality of the paper; indeed I consider it inappropriate; and particularly Figure 1 seems to me not appropriate for a scientific paper. My suggestion is to remove figure 1, this "original" motivation for research, and the relevant comments in the conclusions. In any case, this comment expresses my personal opinion and I leave the final decision about this to the editor.
OK, we will do that.
2) Figure 2: which is the good reason to show graph (b)? It reports data apparently not relevant to the text. Give some valid reasons to show it or remove it!
OK, in fact part b) can be removed
3) Lines 207-215 “As no comprehensive review of the use of niobium catalysts in biomass conversion has been published yet, we would like to fill this gap with presenting a review of the use of niobium catalysts for converting biomass fractions such as cellulose, hemicellulose, and lignin into initial platform molecules for high value-added products. The catalytic conversion of plant biomass lipids by niobium compounds as well as the conversion of glycerol, the byproduct derived from those reactions, into value-added molecules will also be discussed. The proposed review fills the gap in the review papers devoted to niobium catalysts because it focuses on the catalytic reactions (conversion of biomass and biomass derivatives) not covered yet in any review article”
I suggest you review this paragraph, which brings up the same concepts several times. I strongly suggest making it leaner, more concise and clearer, and less redundant.
OK, in fact this paragraph should be modified.
4) The review is organized into two main chapters: Chap 2 and Chap 3. Each chapter is subdivided into paragraphs and sub-paragraphs each dedicated to a reaction of one or two biomass species or derivatives. Being the manuscript is quite long, it appears a bit disorienting for the reader: I suggest adding a short index between paragraph "1.4-Scope of the review" and chapter 2 to improve the consulting facility.
OK
Reviewer 4 Report
This review by Ziolek, Olveira and co-workers delas with recent works about Niobium based catalysis in biomass conversion. The topic is of a certain interest, and the review is fairly comporhensive. The results of the work are presented adequately. My main concern regards the figures and particularly structures, which vary too much in format.
In particular:
1) in Figures 8,9,10,12, 15 and 19 all the structures have different settings. Please redraw the structures using a consistent format.
2) Figure 13 and 21 have odd bond angles. Also, Figure 27 has rectangles superimposed to the structures.
3) Please correct WO3 in WO3 thoroughout the text.
Based on these considerations, I recommend minor revision of the manuscript
Author Response
ND
Round 2
Reviewer 1 Report
Accept